# NCP-VAE: VARIATIONAL AUTOENCODERS WITH NOISE CONTRASTIVE PRIORS

## ABSTRACT

Variational autoencoders (VAEs) are one of the powerful likelihood-based generative models with applications in various domains. However, they struggle to generate high-quality images, especially when samples are obtained from the prior without any tempering. One explanation for VAEs' poor generative quality is the prior hole problem: the prior distribution fails to match the aggregate approximate posterior. Due to this mismatch, there exist areas in the latent space with high density under the prior that do not correspond to any encoded image. Samples from those areas are decoded to corrupted images. To tackle this issue, we propose an energy-based prior defined by the product of a base prior distribution and a reweighting factor, designed to bring the base closer to the aggregate posterior. We train the reweighting factor by noise contrastive estimation, and we generalize it to hierarchical VAEs with many latent variable groups. Our experiments confirm that the proposed noise contrastive priors improve the generative performance of state-of-the-art VAEs by a large margin on the MNIST, CIFAR-10, CelebA 64, and CelebA HQ 256 datasets.

## 1 INTRODUCTION

Variational autoencoders (VAEs) (Kingma & Welling, 2014; Rezende et al., 2014) are one of the powerful likelihood-based generative models that have applications in image generation (Brock et al., 2018; Karras et al., 2019; Razavi et al., 2019), music synthesis (Dhariwal et al., 2020), speech generation (Oord et al., 2016; Ping et al., 2020), image captioning (Aneja et al., 2019; Deshpande et al., 2019; Aneja et al., 2018), semi-supervised learning (Kingma et al., 2014; Izmailov et al., 2020), and representation learning (Van Den Oord et al., 2017; Fortuin et al., 2018). Although there has been tremendous progress in improving the expressivity of the approximate posterior, several studies have observed that VAE priors fail to match the *aggregate (approximate) posterior* (Rosca et al., 2018; Hoffman & Johnson, 2016). This phenomenon is sometimes described as *holes in the prior*, referring to regions in the latent space that are not decoded to data-like samples. Such regions often have a high density under the prior but have a low density under the aggregate approximate posterior.

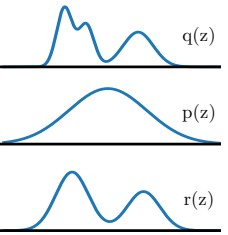

**Figure 1:** We propose an EBM prior using the product of a base prior $p(\mathbf{z})$ and a reweighting factor $r(\mathbf{z})$, designed to bring the base prior closer to the aggregate posterior $q(\mathbf{z})$.

The prior hole problem is commonly tackled by increasing the flexibility of the prior via hierarchical priors (Klushyn et al., 2019), autoregressive models (Gulrajani et al., 2016), a mixture of approximate posteriors (Tomczak & Welling, 2018), normalizing flows (Xu et al., 2019; Chen et al., 2016), resampled priors (Bauer & Mnih, 2019), and energy-based models (Pang et al., 2020; Vahdat et al., 2018b;a; 2020). Among them, energy-based models (EBMs) (Du & Mordatch, 2019; Pang et al., 2020) have shown promising results in learning expressive priors. However, they require running iterative MCMC steps during training which is computationally expensive, especially when the energy function is represented by a

neural network. Moreover, they scale poorly to hierarchical models where an EBM is defined on each group of latent variables.

Our key insight in this paper is that a trainable prior is brought as close as possible to the aggregate posterior as a result of training a VAE. The mismatch between the prior and the aggregate posterior can be reduced by simply reweighting the prior to re-adjust its likelihood in the area of mismatch with the aggregate posterior. To represent this reweighting mechanism, we formulate the prior using an EBM that is defined by the product of a reweighting factor and a base trainable prior as shown in Fig. 1. We represent the reweighting factor using neural networks and the base prior using Normal distributions.

Instead of expensive MCMC sampling, we use noise contrastive estimation (NCE) (Gutmann & Hyvärinen, 2010) for training the EBM prior. We show that NCE naturally trains the reweighting factor in our prior by learning a binary classifier to distinguish samples from a target distribution (i.e., samples from the approximate posterior) vs. samples from a noise distribution (i.e., the base trainable prior). However, since NCE's success depends on how close the noise distribution is to the target distribution, we first train the VAE with the base prior to bring it close to the aggregate posterior. And then, we train the EBM prior using NCE.

In this paper, we make the following contributions: i) We propose an EBM prior termed *noise contrastive prior (NCP)* which is trained by contrasting samples from the aggregate posterior to samples from a base prior. NCPs are learned as a post-training mechanism to replace the original prior with a more flexible prior, which can improve the generative performance of VAEs with any structure. ii) We also show how NCPs are trained on hierarchical VAEs with many latent variable groups. We show that training hierarchical NCPs scales easily to many groups, as they are trained for each latent variable group in parallel. iii) Finally, we demonstrate that NCPs improve the generative quality of VAEs by a large margin across datasets.

## 2 BACKGROUND

We first review VAEs, their extension to hierarchical VAEs, and the prior hole problem.

**Variational Autoencoders:** VAEs learn a generative distribution $p(\mathbf{x}, \mathbf{z}) = p(\mathbf{z})p(\mathbf{x}|\mathbf{z})$ where $p(\mathbf{z})$ is a prior distribution over the latent variable $\mathbf{z}$ and $p(\mathbf{x}|\mathbf{z})$ is a likelihood function that generates the data $\mathbf{x}$ given $\mathbf{z}$. VAEs are trained by maximizing a variational lower bound on the log-likelihood $\log p(\mathbf{x})$:

$$\log p(\mathbf{x}) \geq \mathbb{E}_{\mathbf{z} \sim q(\mathbf{z}|\mathbf{x})}[\log p(\mathbf{x}|\mathbf{z})] - \mathrm{KL}(q(\mathbf{z}|\mathbf{x})||p(\mathbf{z})) := \mathcal{L}_{\mathrm{VAE}}(\mathbf{x}), \tag{1}$$

where $q(\mathbf{z}|\mathbf{x})$ is an approximate posterior and KL is the Kullback–Leibler divergence. The final training objective is formulated by $\mathbb{E}_{p_d(\mathbf{x})}[\mathcal{L}_{\mathrm{VAE}}(\mathbf{x})]$ where $p_d(\mathbf{x})$ is the data distribution (Kingma & Welling, 2014).

**Hierarchical VAEs (HVAEs):** To increase the expressivity of both prior and approximate posterior, earlier work adapted a hierarchical latent variable structure (Vahdat & Kautz, 2020; Kingma et al., 2016; Sønderby et al., 2016; Gregor et al., 2016). In HVAEs, the latent variable $\mathbf{z}$ is divided into $K$ separate *groups*, $\mathbf{z} = \{\mathbf{z}_1, \ldots, \mathbf{z}_K\}$. The approximate posterior and the prior distributions are then defined by $q(\mathbf{z}|\mathbf{x}) = \prod_{k=1}^{K} q(\mathbf{z}_k|\mathbf{z}_{<k}, \mathbf{x})$ and $p(\mathbf{z}) = \prod_{k=1}^{K} p(\mathbf{z}_k|\mathbf{z}_{<k})$. Using these, the training objective becomes:

$$\mathcal{L}_{\mathrm{HVAE}}(\mathbf{x}) := \mathbb{E}_{q(\mathbf{z}|\mathbf{x})}[\log p(\mathbf{x}|\mathbf{z})] - \sum_{k=1}^{K} \mathbb{E}_{q(\mathbf{z}_{<k}|\mathbf{x})}\left[\mathrm{KL}(q(\mathbf{z}_k|\mathbf{z}_{<k}, \mathbf{x})||p(\mathbf{z}_k|\mathbf{z}_{<k}))\right], \tag{2}$$

where $q(\mathbf{z}_{<k}|\mathbf{x}) = \prod_{i=1}^{k-1} q(\mathbf{z}_i|\mathbf{z}_{<i}, \mathbf{x})$ is the approximate posterior up to the $(k-1)^{\mathrm{th}}$ group[1].

**The Prior Hole Problem:** Let $q(\mathbf{z}) \triangleq \mathbb{E}_{p_d(\mathbf{x})}[q(\mathbf{z}|\mathbf{x})]$ denote the aggregate (approximate) posterior. In Appendix B.1, we show that maximizing $\mathbb{E}_{p_d(\mathbf{x})}[\mathcal{L}_{\mathrm{VAE}}(\mathbf{x})]$ with respect to the prior parameters corresponds to bringing the prior as close as possible to the aggregate posterior by minimizing $\mathrm{KL}(q(\mathbf{z})||p(\mathbf{z}))$ w.r.t. $p(\mathbf{z})$. Formally, the prior hole problem refers to the phenomenon that $p(\mathbf{z})$ fails to match $q(\mathbf{z})$.

---

[1] For $k = 1$, the expectation inside the summation is simplified to $\mathrm{KL}(q(\mathbf{z}_1|\mathbf{x})||p(\mathbf{z}_1))$.

## 3 Noise Contrastive Priors (NCPs)

One of the main causes of the prior hole problem is the limited expressivity of the prior that prevents it from matching the aggregate posterior. Recently, energy-based models have shown promising results in representing complex distributions. Motivated by their success, we introduce the noise contrastive prior (NCP) $p_{\text{NCP}}(\mathbf{z}) = \frac{1}{Z} r(\mathbf{z}) p(\mathbf{z})$, where $p(\mathbf{z})$ is a base prior distribution, e.g., a Normal, $r(\mathbf{z})$ is a reweighting factor, and $Z = \int r(\mathbf{z}) p(\mathbf{z}) d\mathbf{z}$ is the normalization constant. The function $r : \mathbb{R}^n \to \mathbb{R}^+$ maps the latent variable $\mathbf{z} \in \mathbb{R}^n$ to a positive scalar, and can be implemented using neural nets.

The reweighting factor $r(\mathbf{z})$ can be trained using MCMC sampling as discussed in Appendix A. However, MCMC requires expensive sampling iterations that scale poorly to hierarchical VAEs. To address this, we describe a noise contrastive estimation based approach to train $p_{\text{NCP}}(\mathbf{z})$ without MCMC sampling.

### 3.1 Learning The Reweighting Factor with Noise Contrastive Estimation

Recall that training VAEs closes the gap between the prior and the aggregate posterior by minimizing $\text{KL}(q(\mathbf{z})||p(\mathbf{z}))$ with respect to prior. Assuming the base prior $p(\mathbf{z})$ to be fixed, $\text{KL}(q(\mathbf{z})||p_{\text{NCP}}(\mathbf{z}))$ is zero when $r(\mathbf{z}) = q(\mathbf{z})/p(\mathbf{z})$. However, since we do not have the density function for $q(\mathbf{z})$, we cannot compute the ratio explicitly. Instead, in this paper, we propose to estimate $r(\mathbf{z})$ using noise contrastive estimation (Gutmann & Hyvärinen, 2010), also known as the likelihood ratio trick that has been popularized in machine learning by predictive coding (Oord et al., 2018) and generative adversarial networks (GANs) (Goodfellow et al., 2014). Since, we can generate samples from both $p(\mathbf{z})$ and $q(\mathbf{z})^2$, we train a binary classifier to distinguish samples from $q(\mathbf{z})$ and samples from the base prior $p(\mathbf{z})$ by minimizing the binary cross-entropy loss:

$$\min_D - \mathbb{E}_{\mathbf{z} \sim q(\mathbf{z})}[\log D(\mathbf{z})] - \mathbb{E}_{\mathbf{z} \sim p(\mathbf{z})}[\log(1 - D(\mathbf{z}))]. \tag{3}$$

Here, $D : \mathbb{R}^n \to (0, 1)$ is a binary classifier that generates the classification prediction probabilities. Eq. (3) is minimized when $D(\mathbf{z}) = \frac{q(\mathbf{z})}{q(\mathbf{z}) + p(\mathbf{z})}$. Denoting the classifier at optimality by $D^*(\mathbf{z})$, we estimate the reweighting factor $r(\mathbf{z}) = \frac{q(\mathbf{z})}{p(\mathbf{z})} \approx \frac{D^*(\mathbf{z})}{1 - D^*(\mathbf{z})}$. The appealing advantage of this estimator is that it is obtained by simply training a binary classifier rather than using expensive MCMC sampling.

### 3.2 Two-stage Training for Noise Contrastive Priors

To properly learn the reweighting factor, NCE training requires the base prior distribution to be close to the target distribution. Intuitively, if $p(\mathbf{z})$ is very close to $q(\mathbf{z})$ (i.e., $p(\mathbf{z}) \approx q(\mathbf{z})$), the optimal classifier will have a large loss value in Eq. (3), and we will have $r(\mathbf{z}) \approx 1$. If $p(\mathbf{z})$ is instead far from $q(\mathbf{z})$, the binary classifier will easily learn to distinguish samples from the two distributions and it will not learn the likelihood ratios correctly. If $p(\mathbf{z})$ is roughly close to $q(\mathbf{z})$, then the binary classifier can learn the ratios.

To ensure that the base prior distribution is close to the target aggregate posterior distribution, we propose a two-stage training algorithm. In the first stage, we train the VAE with only the base prior $p(\mathbf{z})$. From Appendix B.1, we know that at the end of training, $p(\mathbf{z})$ is as close as possible to $q(\mathbf{z})$. In the second stage, we freeze the VAE model including the approximate posterior $q(\mathbf{z}|\mathbf{x})$, the base prior $p(\mathbf{z})$, and the likelihood $p(\mathbf{x}|\mathbf{z})$, and we only train the reweighting factor $r(\mathbf{z})$ using Eq. (3). The second stage can be thought of as replacing the base distribution $p(\mathbf{z})$ with a more expressive distribution of the form $p_{\text{NCP}}(\mathbf{z}) \propto r(\mathbf{z}) p(\mathbf{z})$. Hence, NCP matches the prior to the aggregate posterior $q(\mathbf{z})$ by using the reweighting factors. Note that our proposed method is generic as it only assumes that we can draw samples from $q(\mathbf{z})$ and $p(\mathbf{z})$, which applies to any VAE. Our training is illustrated in Fig. 2.

---

[2]We generate samples from the aggregate posterior $q(\mathbf{z}) = \mathbb{E}_{p_d(\mathbf{x})}[q(\mathbf{z}|\mathbf{x})]$ via ancestral sampling: draw data from the training set ($\mathbf{x} \sim p_d(\mathbf{x})$) and then sample from $\mathbf{z} \sim q(\mathbf{z}|\mathbf{x})$.

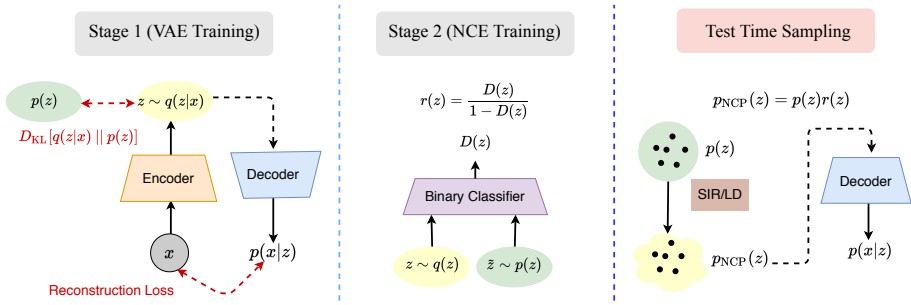

**Figure 2:** NCP-VAE is trained in two stages. In the first stage, we train a VAE using the original VAE objective. In the second stage, we train the reweighting factor $r(\mathbf{z})$ using noise contrastive estimation (NCE). NCE trains a classifier to distinguish samples from the prior and samples from the aggregate posterior. Our noise contrastive prior (NCP) is then constructed by the product of the base prior and the reweighting factor, formed via the classifier. At test time, we sample from NCP using SIR or LD. These samples are then passed to the decoder to generate output samples.

### 3.3 Test Time Sampling

To sample from a VAE with an NCP, we first generate samples from the NCP before passing them to the decoder to generate output samples (shown in Fig. 2). We propose two methods for sampling from NCPs.

**Sampling-Importance-Resampling (SIR):** We first generate $M$ samples from the base prior distribution $\{\mathbf{z}^{(m)}\}_{m=1}^{M} \sim p(\mathbf{z})$. We then resample one of the $M$ proposed samples using importance weights proportional to $w^{(m)} = p_{\mathrm{NCP}}(\mathbf{z}^{(m)})/p(\mathbf{z}^{(m)}) = r(\mathbf{z}^{(m)})$. The benefit of this technique: both proposal generation and the evaluation of $r$ on the samples are done in parallel.

**Langevin Dynamics (LD):** Since our NCP is an EBM, we can use LD for sampling. Denoting the energy function by $E(\mathbf{z}) = -\log r(\mathbf{z}) - \log p(\mathbf{z})$, we initialize a sample $\mathbf{z}_0$ by drawing from $p(\mathbf{z})$ and update the sample iteratively using: $\mathbf{z}_{t+1} = \mathbf{z}_t - 0.5\,\lambda\nabla_{\mathbf{z}}E(\mathbf{z}) + \sqrt{\lambda}\epsilon_t$ where $\epsilon_t \sim \mathcal{N}(0,1)$ and $\lambda$ is the step size. LD is run for a finite number of iterations, and in contrast to SIR, it can be slow given its sequential form.

### 3.4 Generalization to Hierarchical VAEs

The state-of-the-art VAE (Vahdat & Kautz, 2020) uses a hierarchical $q(\mathbf{z}|\mathbf{x})$ and $p(\mathbf{z})$. Here $p(\mathbf{z})$ is chosen to be a Gaussian distribution. Appendix B.2 shows that training a HVAE encourages the prior to minimize $\mathbb{E}_{q(\mathbf{z}_{<k})}\left[\mathrm{KL}(q(\mathbf{z}_k|\mathbf{z}_{<k})||p(\mathbf{z}_k|\mathbf{z}_{<k}))\right]$ for each conditional, where $q(\mathbf{z}_{<k}) \triangleq \mathbb{E}_{p_d(\mathbf{x})}[q(\mathbf{z}_{<K}|\mathbf{x})]$ is the aggregate posterior up to the $(k-1)^{\mathrm{th}}$ group, and $q(\mathbf{z}_k|\mathbf{z}_{<k}) \triangleq \mathbb{E}_{p_d(\mathbf{x})}[q(\mathbf{z}_k|\mathbf{z}_{<k},\mathbf{x})]$ is the aggregate conditional for the $k^{\mathrm{th}}$ group. Given this observation, we extend NCPs to hierarchical models to match each conditional in the prior with $q(\mathbf{z}_k|\mathbf{z}_{<k})$. Formally, we define hierarchical NCPs by $p_{\mathrm{NCP}}(\mathbf{z}) = \frac{1}{Z}\prod_{k=1}^{K} r(\mathbf{z}_k|\mathbf{z}_{<k})p(\mathbf{z}_k|\mathbf{z}_{<k})$ where each factor is an EBM. $p_{\mathrm{NCP}}(\mathbf{z})$ resembles energy machines with an autoregressive structure among groups (Nash & Durkan, 2019).

In the first stage, we train the HVAE with prior $\prod_{k=1}^{K} p(\mathbf{z}_k|\mathbf{z}_{<k})$. For the second stage, we use $K$ binary classifiers, each for a hierarchical group. Following Appendix C, we train each classifier via:

$$\min_{D_k} \mathbb{E}_{p_d(\mathbf{x})q(\mathbf{z}_{<k}|\mathbf{x})}\left[ -\mathbb{E}_{q(\mathbf{z}_k|\mathbf{z}_{<k},\mathbf{x})}[\log D_k(\mathbf{z}_k, c(\mathbf{z}_{<k}))] - \mathbb{E}_{p(\mathbf{z}_k|\mathbf{z}_{<k})}[\log(1 - D_k(\mathbf{z}_k, c(\mathbf{z}_{<k})))] \right], \quad (4)$$

where the outer expectation samples from groups up to the $(k-1)^{\text{th}}$ group, and the inner expectations sample from approximate posterior and base prior for the $k^{\text{th}}$ group, conditioned on the same $\mathbf{z}_{<k}$. The discriminator $D_k$ classifies samples $\mathbf{z}_k$ while conditioning its prediction on $\mathbf{z}_{<k}$ using a shared context feature $c(\mathbf{z}_{<k})$.

The NCE training in Eq. (4) is minimized when $D_k(\mathbf{z}_k, c(\mathbf{z}_{<k})) = \frac{q(\mathbf{z}_k | \mathbf{z}_{<k})}{q(\mathbf{z}_k | \mathbf{z}_{<k}) + p(\mathbf{z}_k | \mathbf{z}_{<k})}$. Denoting the classifier at optimality by $D_k^*(\mathbf{z}, c(\mathbf{z}_{<k}))$, we obtain the reweighting factor $r(\mathbf{z}_k | \mathbf{z}_{<k}) \approx \frac{D_k^*(\mathbf{z}_k, c(\mathbf{z}_{<k}))}{1 - D_k^*(\mathbf{z}_k, c(\mathbf{z}_{<k}))}$ in the second stage. Given our hierarchical NCP, we use ancestral sampling to sample from the prior. For sampling from each group, we can use SIR or LD as discussed before.

The context feature $c(\mathbf{z}_{<k})$ extracts a representation from $\mathbf{z}_{<k}$. Instead of learning a new representation at stage two, we simply use the representation that is extracted from $\mathbf{z}_{<k}$ in the hierarchical prior, trained in the first stage. Note that the binary classifiers in the second stage are trained in parallel for all groups.

## 4 RELATED WORK

In this section, we review prior works related to the proposed method.

**Energy-based Models (EBMs):** Early work on EBMs for generative learning goes back to 1980s (Ackley et al., 1985; Hinton et al., 1986). Prior to the modern deep learning era, most attempts for building generative models using EBMs were centered around Boltzmann machines (Hinton, 2002; Hinton et al., 2006) and their "deep" extensions (Salakhutdinov & Hinton, 2009; Larochelle & Bengio, 2008). Although the energy function in these models is restricted to simple bilinear functions, they have been proven effective for representing the prior in discrete VAEs (Rolfe, 2016; Vahdat et al., 2018a;b; 2020). Recently, EBMs with neural energy functions have gained popularity for representing complex data distribution (Du & Mordatch, 2019). Pang et al. (2020) have shown that neural EBMs can represent expressive prior distributions. However, in this case, the prior is trained using MCMC sampling, and it has been limited to a single group of latent variables. To eliminate MCMC sampling, NCE (Gutmann & Hyvärinen, 2010) has recently been used for training a normalizing flow on data distributions (Gao et al., 2020). Moreover, Han et al. (2019; 2020) have used divergence triangulation to sidesteps MCMC sampling. In contrast, we use NCE to train an EBM prior where a noise distribution is easily available through a pre-trained VAE.

**Adversarial Training:** Similar to NCE, generative adversarial networks (GANs) (Goodfellow et al., 2014) also rely on a discriminator to learn the likelihood ratio between noise and real images. However, GANs use the discriminator to update the generator, whereas in NCE, the noise generator is fixed. In spirit similar are recent works (Azadi et al., 2018; Turner et al., 2019; Che et al., 2020) that link GANs, defined in the pixels space, to EBMs. We apply the likelihood ratio trick to the latent space of VAEs. The main difference: the base prior and approximate posterior are trained with the VAE objective rather than the adversarial loss. Adversarial loss has been used for training implicit encoders in VAEs (Makhzani et al., 2015; Mescheder et al., 2017; Engel et al., 2018). But, they have not been linked to energy-based priors as we do explicitly.

**Prior Hole Problem:** Among prior works on this problem, VampPrior (Tomczak & Welling, 2018) uses a mixture of encoders to represent the prior. However, this requires storing training data or pseudo-data to generate samples at test time. Takahashi et al. (2019) use the likelihood ratio estimator to train a simple prior distribution. However at test time, the aggregate posterior is used for sampling in the latent space. Bauer & Mnih (2019) propose a reweighting factor similar to ours, but it is trained via importance sampling. Recently, Lawson et al. (2019) introduced *energy-inspired models (EIMs)* that define distributions induced by the sampling processes used by (Bauer & Mnih, 2019) as well as our SIR sampling. Although, EIMs have the advantage of end-to-end training and can be used as either prior or decoder in VAEs, they require multiple samples during training (up to 1K). This can make application of EIMs to deep hierarchical models such as NVAEs very challenging as these models are often very memory intensive and are trained with a few training samples per GPU.

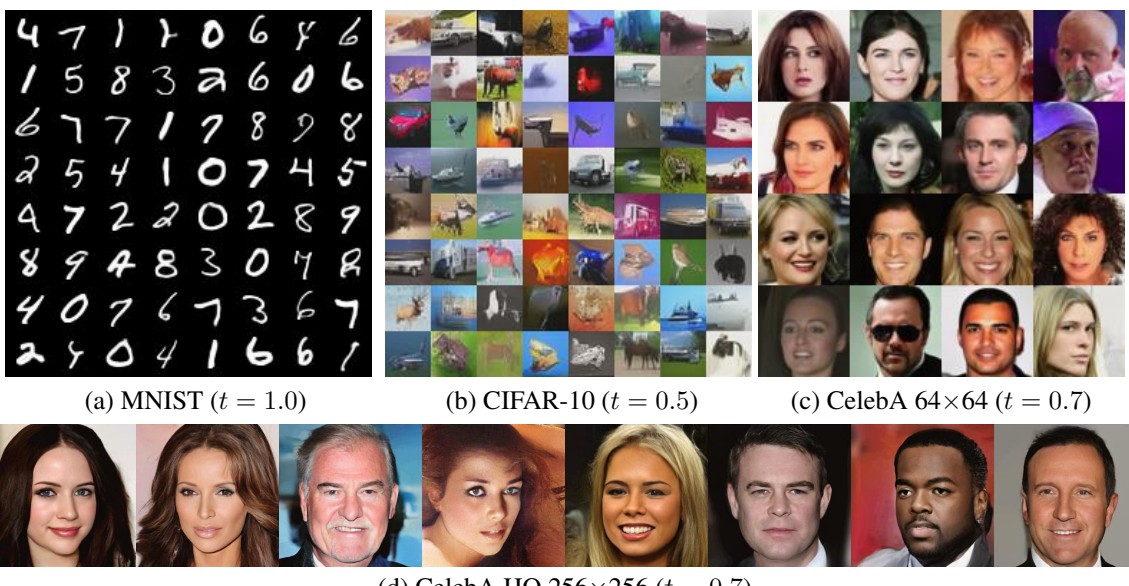

(a) MNIST ($t = 1.0$)       (b) CIFAR-10 ($t = 0.5$)       (c) CelebA 64×64 ($t = 0.7$)

(d) CelebA HQ 256×256 ($t = 0.7$)

**Figure 3:** Randomly sampled images from NCP-VAE with the temperature $t$ for the prior.

**Two-stage VAEs:** VQ-VAE (Van Den Oord et al., 2017; Razavi et al., 2019) first trains an autoencoder and then fits an autoregressive PixelCNN (Van Den Oord et al., 2016) prior to the latent variables which is slow to sample from. Two-stage VAE (2s-VAE) (Dai & Wipf, 2018) trains a VAE on the data, and then, trains another VAE in the latent space. Regularized autoencoders (RAE) (Ghosh et al., 2020) train an autoencoder, and subsequently a Gaussian mixture model on latent codes. In contrast, we train the model with the original VAE objective in the first stage, and we improve the expressivity of the prior using an EBM.

## 5 EXPERIMENTS

Our implementation of NCP-VAE builds upon NVAE (Vahdat & Kautz, 2020), the state-of-the-art hierarchical VAE. We examine NCP-VAE on four datasets including dynamically binarized MNIST (LeCun, 1998), CIFAR-10 (Krizhevsky et al., 2009), CelebA-64 (Liu et al., 2015) and CelebA-HQ-256 (Karras et al., 2017). For CIFAR-10 and CelebA-64, the model has 30 groups, and for CelebA-HQ-256 it has 20 groups. We sample from NCP-VAE using SIR with 5K proposal samples at the test (image generation) time. On these datasets, we measure the sample quality using the Fréchet Inception Distance (FID) score (Heusel et al., 2017) with 50,000 samples, as computing the log-likelihood requires estimating the intractable normalization constant. To report log-likelihood results, we train an NVAE model with a small latent space on MNIST with 10 groups of $4 \times 4$ latent variables. We estimate log normalization constant in NCPs using 1000 importance weighted samples. We intentionally limit the latent space to ensure that we can estimate the normalization constant correctly (standard deviation of $\log Z$ estimation $\leq 0.23$). Thus, on this dataset, we report negative log-likelihood (NLL). Implementation details are provided in Appendix E.

### 5.1 QUANTITATIVE RESULTS

The quantitative results are reported in Tab. 1, Tab. 2, Tab. 3, and Tab. 4. On all four datasets, our model improves upon state-of-the-art NVAE, and it reduces the gap with GANs by a large margin. On CelebA 64, we improve NVAE from an FID of 13.48 to 5.25, comparable to GANs. On CIFAR-10, NCP-VAE

**Table 1:** Generative performance on CelebA-64

| Model | FID↓ |
|---|---|
| NCP-VAE (ours) | **5.25** |
| NVAE (Vahdat & Kautz, 2020) | 13.48 |
| RAE (Ghosh et al., 2020) | 40.95 |
| 2s-VAE (Dai & Wipf, 2018) | 44.4 |
| WAE (Tolstikhin et al., 2018) | 35 |
| Perceptial AE(Zhang et al., 2020) | 13.8 |
| Latent EBM (Pang et al., 2020) | 37.87 |
| COCO-GAN (Lin et al., 2019) | 4.0 |
| QA-GAN (Parimala & Channappayya, 2019) | 6.42 |
| NVAE-Recon (Vahdat & Kautz, 2020) | 1.03 |

**Table 2:** Generative performance on CIFAR-10

| Model | FID↓ |
|---|---|
| NCP-VAE (ours) | **24.08** |
| NVAE (Vahdat & Kautz, 2020) | 51.71 |
| RAE (Ghosh et al., 2020) | 74.16 |
| 2s-VAE (Dai & Wipf, 2018) | 72.9 |
| Perceptial AE (Zhang et al., 2020) | 51.51 |
| EBM (Du & Mordatch, 2019) | 40.58 |
| Latent EBM (Pang et al., 2020) | 70.15 |
| Style-GANv2 (Karras et al., 2020) | 3.26 |
| Denoising Diffusion Process (Ho et al., 2020) | 3.17 |
| NVAE-Recon (Vahdat & Kautz, 2020) | 2.67 |

**Table 3:** Generative results on CelebA-HQ-256

| Model | FID↓ |
|---|---|
| NCP-VAE (ours) | **24.79** |
| NVAE (Vahdat & Kautz, 2020) | 40.26 |
| GLOW (Kingma & Dhariwal, 2018) | 68.93 |
| Advers. LAE (Pidhorskyi et al., 2020) | 19.21 |
| PGGAN (Karras et al., 2017) | 8.03 |
| NVAE-Recon (Vahdat & Kautz, 2020) | 0.45 |

**Table 4:** Likelihood results on MNIST in nats

| Model | NLL↓ |
|---|---|
| NCP-VAE (ours) | **78.10** |
| NVAE-small (Vahdat & Kautz, 2020) | 78.67 |
| BIVA (Maaløe et al., 2019) | 78.41 |
| DAVE++ (Vahdat et al., 2018b) | 78.49 |
| IAF-VAE (Kingma et al., 2016) | 79.10 |
| VampPrior AR dec. (Tomczak & Welling) | 78.45 |

improves the NVAE FID of 51.71 to 24.08. On MNIST, although our latent space is much smaller, our model outperforms previous VAEs. NVAE has reported 78.01 nats on this dataset with a larger latent space.

**Reconstruction:** In the last row of Tab. 1, Tab. 2, and Tab. 3, we report the FID score for the reconstructed images of the NVAE baseline. Note how reconstruction FID is much lower than our FID, indicating that our model is far from memorizing the training data. In Appendix G, we also provide nearest neighbours from training data for generated samples. In Appendix F, we present our NCP applied to vanilla VAEs.

## 5.2 QUALITATIVE RESULTS

We visualize samples generated by NCP-VAE in Fig. 3 without any manual intervention. We adopt the common practice of reducing the temperature of the base prior $p(\mathbf{z})$ by scaling down the standard-deviation of the conditional Normal distributions (Kingma & Dhariwal, 2018)[3]. Brock et al. (2018); Vahdat & Kautz (2020) also observe that re-adjusting the batch-normalization (BN), given a temperature applied to the prior, improves the generative quality. Similarly, we achieve diverse, high-quality images by re-adjusting the BN statistics as described by Vahdat & Kautz (2020). Additional qualitative results are shown in Appendix H.

## 5.3 ADDITIONAL STUDIES

We perform additional experiments to study i) how hierarchical NCPs perform as the number of latent groups increases, ii) the impact of SIR and LD hyperparameters, and iii) what the classification loss in NCE training conveys about $p(\mathbf{z})$ and $q(\mathbf{z})$. All experiments in this section are performed on CelebA-64 data.

---

[3]Lowering of the temperature is only used to obtain qualitative samples. It's not used when computing any of the quantitative results in Sec. 5.1.

**Table 6:** Effect of SIR sample size and LD iterations. Time-$N$ is the time used to generate a batch of $N$ images.

| # SIR proposal samples | FID↓ | Time-1 (sec) | Time-10 (sec) | Memory (GB) | # LD iterations | FID↓ | Time-1 (sec) | Time-10 (sec) | Memory (GB) |
|---|---|---|---|---|---|---|---|---|---|
| 5 | 11.75 | 0.34 | 0.42 | 1.96 | 5 | 14.44 | 3.08 | 3.07 | 1.94 |
| 50 | 8.58 | 0.40 | 1.21 | 4.30 | 50 | 12.76 | 27.85 | 28.55 | 1.94 |
| 500 | 6.76 | 1.25 | 9.43 | 20.53 | 500 | 8.12 | 276.13 | 260.35 | 1.94 |
| 5000 | **5.25** | 10.11 | 95.67 | 23.43 | 1000 | **6.98** | 552 | 561.44 | 1.94 |

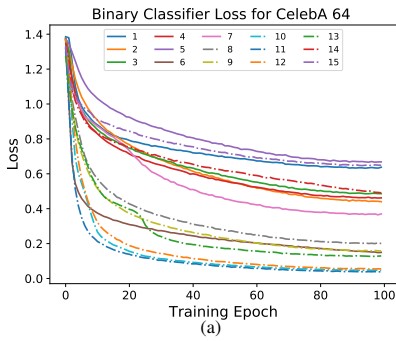 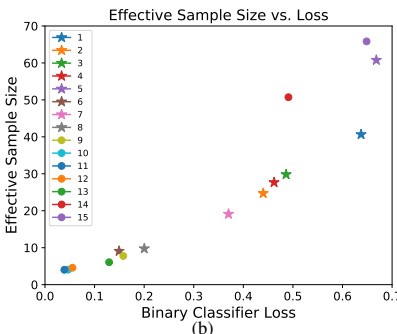

**Figure 4: (a)** Classification loss for binary classifiers on latent variable groups. A larger final loss upon training indicates that $q(\mathbf{z})$ and $p(\mathbf{z})$ are more similar. **(b)** The effective sample size vs. the final loss value at the end of training. Higher effective sample size implies similarity of two distributions.

**Number of latent variable groups:** Tab. 5 shows the generative performance of hierarchical NCP with different amounts of latent variable groups. As we increase the number of groups, the FID score of both NVAE and our model improves. This demonstrates the efficacy of our NCPs, even with expressive hierarchical priors and in the presence of many groups.

**Table 5:** # groups & generative performance in FID↓

| # groups | NVAE | NCP-VAE |
|---|---|---|
| 6 | 33.18 | 18.68 |
| 15 | 14.96 | 5.96 |
| 30 | 13.48 | **5.25** |

**SIR and LD parameters:** The computationally complexity of SIR is similar to LD if we set the number of proposal samples in SIR equal to the number LD iterations. Tab. 6 reports the impact of these parameters. We observe that increasing both the number of proposal samples in SIR and the LD iterations leads to a noticeable improvement in FID score. For SIR, the proposal generation and the evaluation of $r(\mathbf{z})$ are parallelizable. Hence, as shown in Tab. 6, image generation is faster with SIR than with LD (LD iterations are sequential). However, GPU memory usage scales with the number of SIR proposals, but not with the number of LD iterations. Interestingly, SIR, albeit simple, performs better than LD when using about the same compute.

**Classification loss in NCE:** We can draw a direct connection between the classification loss in Eq. 3 and the similarity of $p(\mathbf{z})$ and $q(\mathbf{z})$. Denoting the classification loss in Eq. 3 at optimality by $\mathcal{L}^*$, Goodfellow et al. (2014) show that $\mathrm{JSD}(p(\mathbf{z})||q(\mathbf{z})) = \log 2 - 0.5 \times \mathcal{L}^*$ where JSD denotes the Jensen–Shannon divergence between two distributions. Fig. 4(a) plots the classification loss (Eq. 4) for each classifier for a 15-group NCP trained on the CelebA-64 dataset. Assume that the classifier loss at the end of training is a good approximation of $\mathcal{L}^*$. We observe that 8 out of 15 groups have $\mathcal{L}^* \geq 0.4$, indicating a good overlap between $p(\mathbf{z})$ and $q(\mathbf{z})$ for those groups. To further assess the impact of the distribution match on SIR sampling, in Fig. 4(b), we visualize the effective sample size (ESS)[4] in SIR vs. $\mathcal{L}^*$ for the same group. We observe a strong correlation between $\mathcal{L}^*$ and the effective sample size. SIR is more reliable on the same 8 groups that have high classification loss. These groups are mostly at the top of the NVAE hierarchy which have been shown to control the global structure of generated samples (see B.6 in Vahdat & Kautz (2020)).

---

[4]ESS measures reliability of SIR via $1/\sum_m (\hat{w}^{(m)})^2$, where $\hat{w}^{(m)} = r(\mathbf{z}^{(m)})/\sum_{m'} r(\mathbf{z}^{(m')})$ (Owen, 2013).

## 5.4 ADDITIONAL COMPARISON TO PRIOR ART

In our previous experiments, we rely on NVAE as the VAE backbone. Although this shows that our approach can be applied to large scale models successfully, this may make comparison against the prior art unfair as they are often applied to smaller models. To provide a fair comparison, in this section, we apply NCP to several commonly used small VAE models.

**Comparison against RAE, 2s-VAE, and WAE:** In Tab. 7 we show the generative performance of our approach applied to the VAE architecture in RAE (Ghosh et al., 2020). Note that this VAE architecture has only one latent variable group. The same base architecture was used in the implementation of 2s-VAE (Dai & Wipf, 2018) and WAE (Tolstikhin et al., 2018). We borrow the training setup from Ghosh et al. (2020) on the CelebA-64 dataset, and compare to the baselines reported in this work. We apply our NCP-VAE on top of vanilla VAE with a Gaussian prior as well as a 10-component Gaussian mixture model (GMM) prior that was proposed in RAEs. Our NCP-VAE improves the performance on the base VAE, improving the FID score to 41.28 from 48.12. Additionally, when NCP is applied to the VAE with GMM prior (the RAE model), it improves its performance from 40.95 to the FID score of 39.00.

**Comparison against LARS and SNIS:** In Tab. 8, we compare NCP-VAE to LARS (Bauer & Mnih, 2019) and SNIS (Lawson et al., 2019) priors on MNIST. We implement our NCP-VAE using the VAE and energy-function architectures that were used by Lawson et al. (2019). We closely follow the training hyperparameters used by Lawson et al. (2019) as well as their approach for obtaining a lower bound on the log likelihood. NCP-VAE obtains the negative log-likelihood (NLL) of 82.82, comparable to Lawson et al. (2019), while outperforming LARS (Bauer & Mnih, 2019).

**Table 7:** Generative performance on CelebA-64 with the RAE (Ghosh et al., 2020) architecture

| Model | FID↓ |
|---|---|
| VAE w/ Gaussian prior | 48.12 |
| 2s-VAE (Dai & Wipf, 2018) | 49.70 |
| WAE (Tolstikhin et al., 2018) | 42.73 |
| RAE (Ghosh et al., 2020) | 40.95 |
| NCP w/ Gaussian prior as base | 41.28 |
| NCP w/ GMM prior as base | **39.00** |

**Table 8:** Likelihood results on MNIST on single latent group model with architecture from LARS (Bauer & Mnih, 2019) & SNIS (Lawson et al., 2019) (results in nats)

| Model | NLL↓ |
|---|---|
| VAE w/ Gaussian prior | 84.82 |
| VAE w/ LARS prior (Bauer & Mnih, 2019) | 83.03 |
| VAE w/ SNIS prior (Lawson et al., 2019) | **82.52** |
| NCP-VAE | 82.82 |

## 6 CONCLUSIONS

The prior hole problem is one of the main reasons for VAEs' poor generative quality. In this paper, we tackled this problem by introducing the noise contrastive prior (NCP), defined by the product of a reweighting factor and a base prior. We showed how the reweighting factor is trained by contrasting samples from the aggregate posterior with samples from the base prior. Our proposal is simple and can be applied to any VAE to increase its prior's expressivity. We also showed how NCP training scales to large hierarchical VAEs, as it can be done in parallel simultaneously for all the groups. Finally, we demonstrated that NCPs improve the generative performance of state-of-the-art NVAEs by a large margin, closing the gap to GANs.

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

## A   TRAINING ENERGY-BASED PRIORS USING MCMC

In this section, we show how a VAE with energy-based model in its prior can be trained. Assuming that the prior is in the form $p_{\text{EBM}}(\mathbf{z}) = \frac{1}{Z} r(\mathbf{z}) p(\mathbf{z})$, the variational bound is of the form:

$$\mathbb{E}_{p_d(\mathbf{x})}[\mathcal{L}_{\text{VAE}}] = \mathbb{E}_{p_d(\mathbf{x})}\left[\mathbb{E}_{q(\mathbf{z}|\mathbf{x})}[\log p(\mathbf{x}|\mathbf{z})] - \text{KL}(q(\mathbf{z}|\mathbf{x})||p_{\text{EBM}}(\mathbf{z}))\right]$$
$$= E_{p_d(\mathbf{x})}\left[\mathbb{E}_{q(\mathbf{z}|\mathbf{x})}[\log p(\mathbf{x}|\mathbf{z}) - \log q(\mathbf{z}|\mathbf{x}) + \log r(\mathbf{z}) + \log p(\mathbf{z})]\right] - \log Z,$$

where the expectation term, similar to VAEs, can be trained using the reparameterization trick. The only problematic term is the log-normalization constant $\log Z$, which captures the gradient with respect to the parameters of the prior $p_{\text{EBM}}(\mathbf{z})$. Denoting these parameters by $\theta$, the gradient of $\log Z$ is obtained by:

$$\frac{\partial}{\partial \theta} \log Z = \frac{1}{Z} \int \frac{\partial (r(\mathbf{z}) p(\mathbf{z}))}{\partial \theta} d\mathbf{z} = \int \frac{r(\mathbf{z}) p(\mathbf{z})}{Z} \frac{\partial \log(r(\mathbf{z}) p(\mathbf{z}))}{\partial \theta} d\mathbf{z} = \mathbb{E}_{P_{EBM}(\mathbf{z})}\left[\frac{\partial \log(r(\mathbf{z}) p(\mathbf{z}))}{\partial \theta}\right], \quad (5)$$

where the expectation can be estimated using MCMC sampling from the EBM prior.

## B   MAXIMIZING THE VARIATIONAL BOUND FROM THE PRIOR'S PERSPECTIVE

In this section, we discuss how maximizing the variational bound in VAEs from the prior's perspective corresponds to minimizing a KL divergence from the aggregate posterior to the prior. Note that this relation has been explored by Hoffman & Johnson (2016); Rezende & Viola (2018); Tomczak & Welling (2018) and we include it here for completeness.

### B.1   VAE WITH A SINGLE GROUP OF LATENT VARIABLES

Denote the aggregate (approximate) posterior by $q(\mathbf{z}) \triangleq \mathbb{E}_{p_d(\mathbf{x})}[q(\mathbf{z}|\mathbf{x})]$. Here, we show that maximizing the $\mathbb{E}_{p_d(\mathbf{x})}[\mathcal{L}_{\text{VAE}}(\mathbf{x})]$ with respect to the prior parameters corresponds to learning the prior by minimizing $\text{KL}(q(\mathbf{z})||p(\mathbf{z}))$. To see this, note that the prior $p(\mathbf{z})$ only participates in the KL term in $\mathcal{L}_{\text{VAE}}$ (Eq. 1). We hence have:

$$\arg\max_{p(\mathbf{z})} \mathbb{E}_{p_d(\mathbf{x})}[\mathcal{L}_{\text{VAE}}(\mathbf{x})] = \arg\min_{p(\mathbf{z})} \mathbb{E}_{p_d(\mathbf{x})}[\text{KL}(q(\mathbf{z}|\mathbf{x})||p(\mathbf{z}))]$$
$$= \arg\min_{p(\mathbf{z})} -\mathbb{E}_{p_d(\mathbf{x})}[H(q(\mathbf{z}|\mathbf{x}))] - \mathbb{E}_{q(\mathbf{z})}[\log p(\mathbf{z})]$$
$$= \arg\min_{p(\mathbf{z})} -H(q(\mathbf{z})) - \mathbb{E}_{q(\mathbf{z})}[\log p(\mathbf{z})]$$
$$= \arg\min_{p(\mathbf{z})} \text{KL}(q(\mathbf{z})||p(\mathbf{z})),$$

where $H(.)$ denotes the entropy. Above, we replaced the expected entropy $\mathbb{E}_{p_d(\mathbf{x})}[H(q(\mathbf{z}|\mathbf{x}))]$ with $H(q(\mathbf{z}))$ as the minimization is with respect to the parameters of the prior $p(\mathbf{z})$.

### B.2   HIERARCHICAL VAEs

Denote hierarchical approximate posterior and prior distributions by: $q(\mathbf{z}|\mathbf{x}) = \prod_{k=1}^{K} q(\mathbf{z}_k|\mathbf{z}_{<k}, \mathbf{x})$ and $p(\mathbf{z}) = \prod_{k=1}^{K} p(\mathbf{z}_k|\mathbf{z}_{<k})$. The hierarchical VAE objective becomes:

$$\mathcal{L}_{\text{HVAE}}(\mathbf{x}) = \mathbb{E}_{q(\mathbf{z}|\mathbf{x})}[\log p(\mathbf{x}|\mathbf{z})] - \sum_{k=1}^{K} \mathbb{E}_{q(\mathbf{z}_{<k}|\mathbf{x})}\left[\text{KL}(q(\mathbf{z}_k|\mathbf{z}_{<k}, \mathbf{x})||p(\mathbf{z}_k|\mathbf{z}_{<k}))\right], \quad (6)$$

where $q(\mathbf{z}_{<k}|\mathbf{x}) = \prod_{i=1}^{k-1} q(\mathbf{z}_i|\mathbf{z}_{<i}, \mathbf{x})$ is the approximate posterior up to the $(k-1)^{\text{th}}$ group. Denote the aggregate posterior up to the $(K-1)^{\text{th}}$ group by $q(\mathbf{z}_{<k}) \triangleq \mathbb{E}_{p_d(\mathbf{x})}[q(\mathbf{z}_{<K}|\mathbf{x})]$ and the aggregate conditional for the $k^{\text{th}}$ group given the previous groups $q(\mathbf{z}_k|\mathbf{z}_{<k}) \triangleq \mathbb{E}_{p_d(\mathbf{x})}[q(\mathbf{z}_k|\mathbf{z}_{<k}, \mathbf{x})]$.

Here, we show that maximizing $\mathbb{E}_{p_d(\mathbf{x})}[\mathcal{L}_{\text{HVAE}}(\mathbf{x})]$ with respect to the prior corresponds to learning the prior by minimizing $\mathbb{E}_{q(\mathbf{z}_{<k})}[\text{KL}(q(\mathbf{z}_k|\mathbf{z}_{<k})||p(\mathbf{z}_k|\mathbf{z}_{<k}))]$ for each conditional:

$$
\begin{aligned}
\underset{p(\mathbf{z}_k|\mathbf{z}_{<k})}{\arg\max} \, \mathbb{E}_{p_d(\mathbf{x})}[\mathcal{L}_{\text{HVAE}}(\mathbf{x})] &= \underset{p(\mathbf{z}_k|\mathbf{z}_{<k})}{\arg\min} \, \mathbb{E}_{p_d(\mathbf{x})} \left[ \mathbb{E}_{q(\mathbf{z}_{<k}|\mathbf{x})} \left[ \text{KL}(q(\mathbf{z}_k|\mathbf{z}_{<k}, \mathbf{x})||p(\mathbf{z}_k|\mathbf{z}_{<k})) \right] \right] \\
&= \underset{p(\mathbf{z}_k|\mathbf{z}_{<k})}{\arg\min} \, -\mathbb{E}_{p_d(\mathbf{x})q(\mathbf{z}_{<k}|\mathbf{x})q(\mathbf{z}_k|\mathbf{z}_{<k}, \mathbf{x})} \left[ \log p(\mathbf{z}_k|\mathbf{z}_{<k}) \right] \\
&= \underset{p(\mathbf{z}_k|\mathbf{z}_{<k})}{\arg\min} \, -\mathbb{E}_{q(\mathbf{z}_k, \mathbf{z}_{<k})} \left[ \log p(\mathbf{z}_k|\mathbf{z}_{<k}) \right] \\
&= \underset{p(\mathbf{z}_k|\mathbf{z}_{<k})}{\arg\min} \, -\mathbb{E}_{q(\mathbf{z}_{<k})} \left[ \mathbb{E}_{q(\mathbf{z}_k|\mathbf{z}_{<k})} \left[ \log p(\mathbf{z}_k|\mathbf{z}_{<k}) \right] \right] \\
&= \underset{p(\mathbf{z}_k|\mathbf{z}_{<k})}{\arg\min} \, \mathbb{E}_{q(\mathbf{z}_{<k})} \left[ -H(q(\mathbf{z}_k|\mathbf{z}_{<k})) - \mathbb{E}_{q(\mathbf{z}_k|\mathbf{z}_{<k})} \left[ \log p(\mathbf{z}_k|\mathbf{z}_{<k}) \right] \right] \\
&= \underset{p(\mathbf{z}_k|\mathbf{z}_{<k})}{\arg\min} \, \mathbb{E}_{q(\mathbf{z}_{<k})} \left[ \text{KL}(q(\mathbf{z}_k|\mathbf{z}_{<k})||p(\mathbf{z}_k|\mathbf{z}_{<k})) \right].
\end{aligned}
\tag{7}
$$

## C  CONDITIONAL NCE FOR HIERARCHICAL VAES

In this section, we describe how we derive the NCE training objective for hierarchical VAEs given in Eq. (4). Our goal is to learn the likelihood ratio between the aggregate conditional $q(\mathbf{z}_k|\mathbf{z}_{<k})$ and the prior $p(\mathbf{z}_k|\mathbf{z}_{<k})$. We can define the NCE objective to train the discriminator $D_k(\mathbf{z}_k, \mathbf{z}_{<k})$ that classifies $\mathbf{z}_k$ given samples from the previous groups $\mathbf{z}_{<k}$ using:

$$
\min_{D_k} \, - \mathbb{E}_{q(\mathbf{z}_k|\mathbf{z}_{<k})}[\log D_k(\mathbf{z}_k, \mathbf{z}_{<k})] - \mathbb{E}_{p(\mathbf{z}_k|\mathbf{z}_{<k})}[\log(1 - D_k(\mathbf{z}_k, \mathbf{z}_{<k}))] \quad \forall \mathbf{z}_{<k}.
\tag{8}
$$

Since $\mathbf{z}_{<k}$ is in a high dimensional space, we cannot apply the minimization $\forall \mathbf{z}_{<k}$. Instead, we sample from $\mathbf{z}_{<k}$ using the aggregate approximate posterior $q(\mathbf{z}_{<k})$ as done for the KL in a hierarchical model (Eq. (7)):

$$
\min_{D_k} \, \mathbb{E}_{q(\mathbf{z}_{<k})} \left[ - \mathbb{E}_{q(\mathbf{z}_k|\mathbf{z}_{<k})}[\log D_k(\mathbf{z}_k, \mathbf{z}_{<k})] - \mathbb{E}_{p(\mathbf{z}_k|\mathbf{z}_{<k})}[\log(1 - D_k(\mathbf{z}_k, \mathbf{z}_{<k}))] \right].
\tag{9}
$$

Since $q(\mathbf{z}_{<k})q(\mathbf{z}_k|\mathbf{z}_{<k}) = q(\mathbf{z}_k, \mathbf{z}_{<k}) = \mathbb{E}_{p_d(\mathbf{x})}[q(\mathbf{z}_{<k}|\mathbf{x})q(\mathbf{z}_k|\mathbf{z}_{<k}, \mathbf{x})]$, we have:

$$
\min_{D_k} \, \mathbb{E}_{p_d(\mathbf{x})q(\mathbf{z}_{<k}|\mathbf{x})} \left[ - \mathbb{E}_{q(\mathbf{z}_k|\mathbf{z}_{<k}, \mathbf{x})}[\log D_k(\mathbf{z}_k, \mathbf{z}_{<k})] - \mathbb{E}_{p(\mathbf{z}_k|\mathbf{z}_{<k})}[\log(1 - D_k(\mathbf{z}_k, \mathbf{z}_{<k}))] \right].
\tag{10}
$$

Finally, instead of passing all the samples from the previous latent variables groups to $D$, we can pass the context feature $c(\mathbf{z}_{<k})$ that extracts a representation from all the previous groups:

$$
\min_{D_k} \, \mathbb{E}_{p_d(\mathbf{x})q(\mathbf{z}_{<k}|\mathbf{x})} \left[ - \mathbb{E}_{q(\mathbf{z}_k|\mathbf{z}_{<k}, \mathbf{x})}[\log D_k(\mathbf{z}_k, c(\mathbf{z}_{<k}))] - \mathbb{E}_{p(\mathbf{z}_k|\mathbf{z}_{<k})}[\log(1 - D_k(\mathbf{z}_k, c(\mathbf{z}_{<k})))] \right].
\tag{11}
$$

## D  NVAE BASED MODEL AND CONTEXT FEATURE

**Context Feature:** The base model NVAE (Vahdat & Kautz, 2020) is hierarchical. To encode the information from the lower levels of the hierarchy to the higher levels, during training of the binary classifiers, we concatenate the context feature $c(\mathbf{z}_{<k})$ to the samples from both $p(\mathbf{z})$ and $q(\mathbf{z})$. The context feature for each group is the output of the residual cell of the top-down model and encodes a representation from $\mathbf{z}_{<k}$.

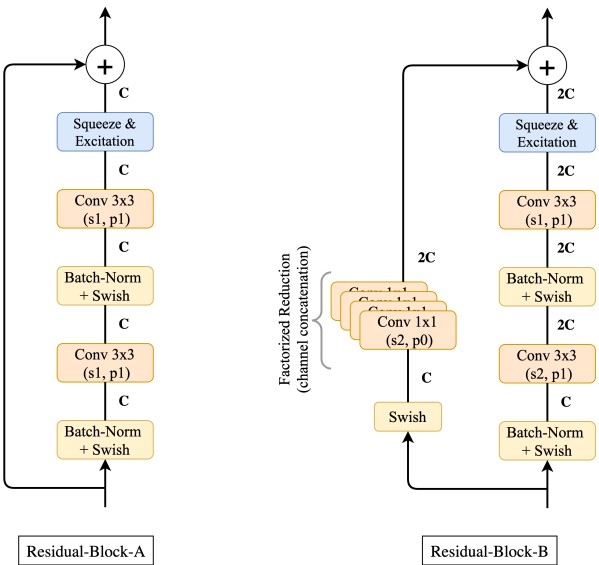

**Figure 5:** Residual blocks used in the binary classifier. We use *s*, *p* and *C* to refer to the stride parameter, the padding parameter and the number of channels in the feature map, respectively.

**Image Decoder** $p(\mathbf{x}|\mathbf{z})$: The base NVAE (Vahdat & Kautz, 2020) uses a mixture of discretized logistic distributions for all the datasets but MNIST, for which it uses a Bernoulli distribution. In our model, we observe that replacing this with a Normal distribution for the RGB image datasets leads to significant improvements in the base model performance. This is also reflected in the gains of our approach.

## E   IMPLEMENTATION DETAILS

The binary classifier is composed of two types of residual blocks as in Fig. 5. The residual blocks use batch-normalization (Ioffe & Szegedy, 2015), the Swish activation function (Ramachandran et al., 2017), and the Squeeze-and-Excitation (SE) block (Hu et al., 2018). SE performs a *squeeze* operation (*e.g.*, mean) to obtain a single value for each channel. An *excitation* operation (non-linear transformation) is applied to these values to get per-channel weights. The Residual-Block-B differs from Residual-Block-A in that it doubles the number of channels ($C \rightarrow 2C$), while down-sampling the other spatial dimensions. It therefore also includes a factorized reduction with $1 \times 1$ convolutions along the skip-connection. The complete architecture of the classifier is:

## F   PERFORMANCE ON SINGLE GROUP VAE

To demonstrate the efficacy of our approach on any off-the-shelf VAE, we, apply our NCE based approach to the VAE in (Ghosh et al., 2020). Note we use the vanilla VAE with Normal prior provided by the authors. The FID for CelebA 64 improves from 48.12 to 41.28. Note that the FID for reconstruction is reported as 39.12. Single group VAEs are known to perform poorly on the task of image geeration, which is reflected in the high FID value of reconstruction.

## G   NEAREST NEIGHBORS FROM THE TRAINING DATASET

To highlight that hierarchical NCP generates unseen samples at test time rather than memorizing the training dataset, Figures 6-7 visualize samples from the model along with a few training images that are most similar

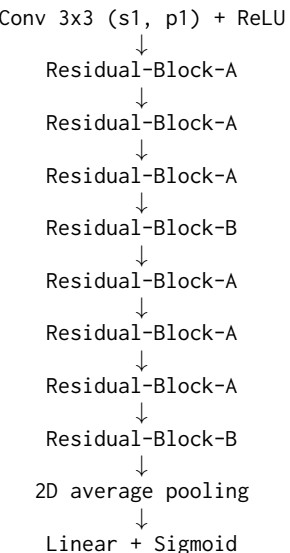

```
Conv 3x3 (s1, p1) + ReLU
           ↓
   Residual-Block-A
           ↓
   Residual-Block-A
           ↓
   Residual-Block-A
           ↓
   Residual-Block-B
           ↓
   Residual-Block-A
           ↓
   Residual-Block-A
           ↓
   Residual-Block-A
           ↓
   Residual-Block-B
           ↓
   2D average pooling
           ↓
   Linear + Sigmoid
```

| | |
|---|---|
| Optimizer | Adam (Kingma & Ba, 2014) |
| Learning Rate | Initialize at $1e$-3, CosineAnnealing (Loshchilov & Hutter, 2016) to $1e$-7 |
| Batch size | 512 (MNIST, CIFAR-10), 256 (CelebA-64), 128 (CelebA HQ 256 ) |

**Table 9:** Hyper-parameters for training the binary classifiers.

to them (nearest neighbors). To get the similarity score for a pair of images, we downsample to $64 \times 64$, center crop to $40 \times 40$ and compute the Euclidean distance. The KD-tree algorithm is used to fetch the nearest neighbors. We note that the generated samples are quite distinct from the training images.

Query Image                                    Nearest neighbors from the training dataset                                    .

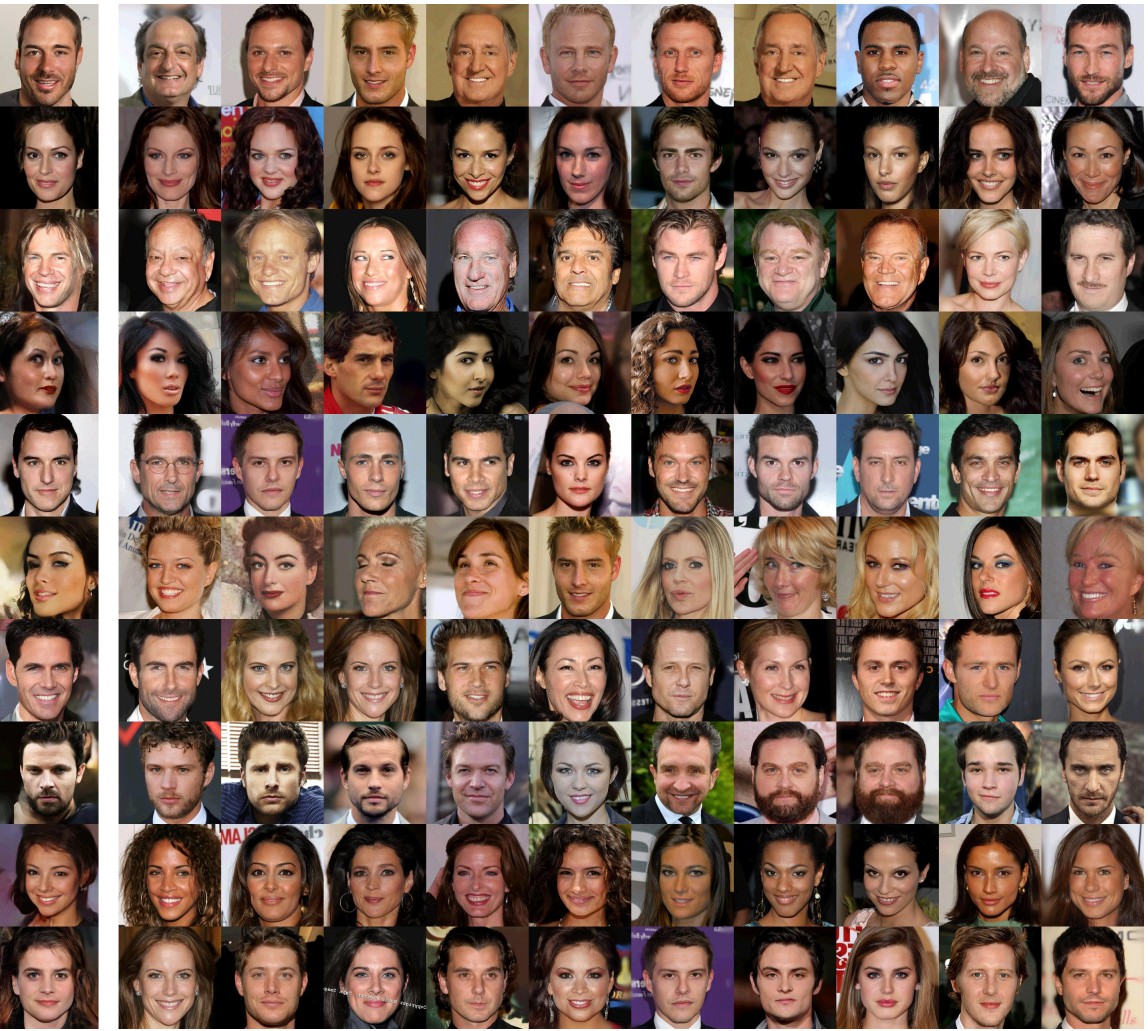

**Figure 6:** Query images (left) and their nearest neighbors from the CelebA-HQ-256 training dataset.

Query Image                          Nearest neighbors from the training dataset                          .

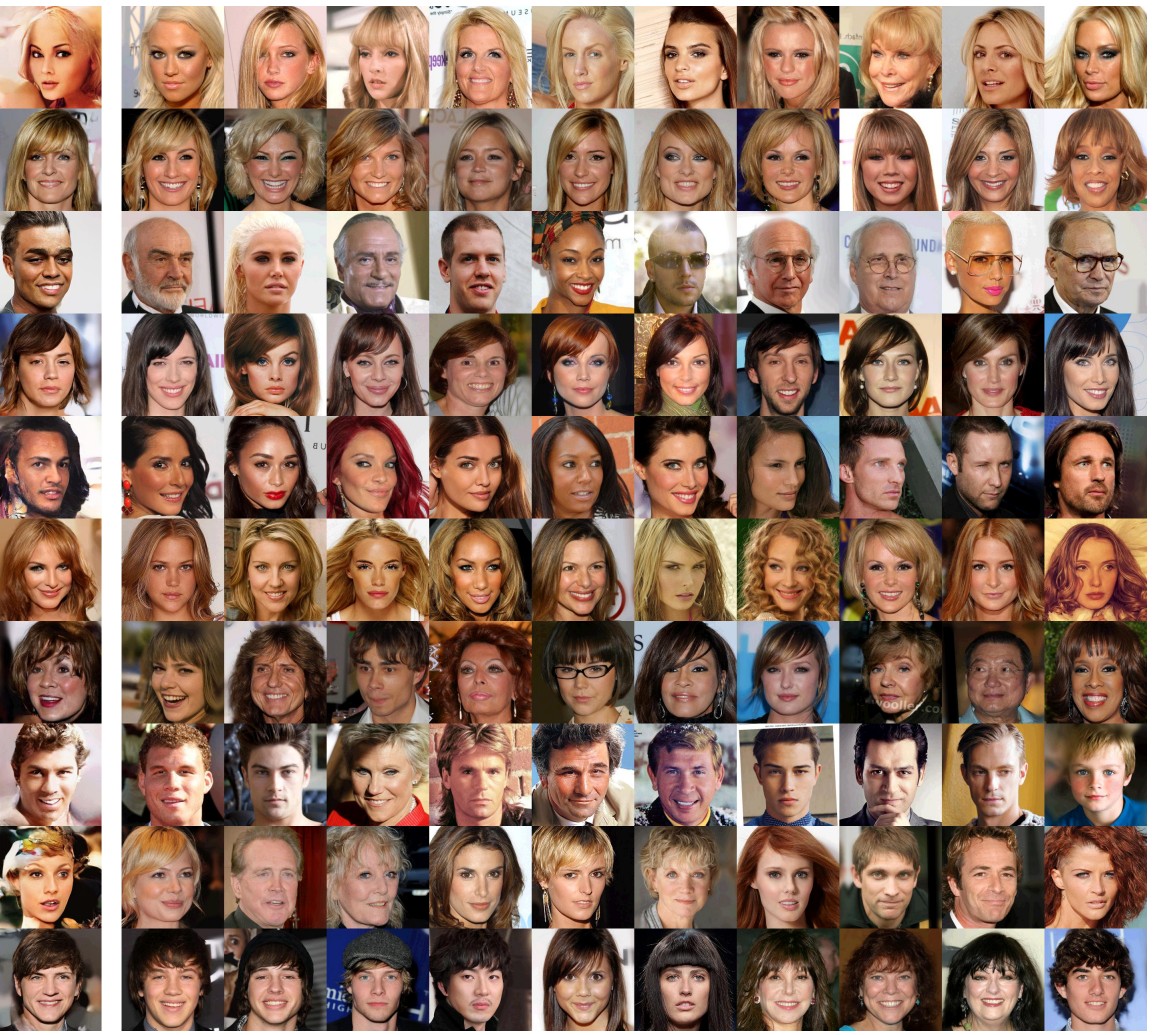

**Figure 7:** Query images (left) and their nearest neighbors from the CelebA-HQ-256 training dataset.

# H    ADDITIONAL QUALITATIVE EXAMPLES

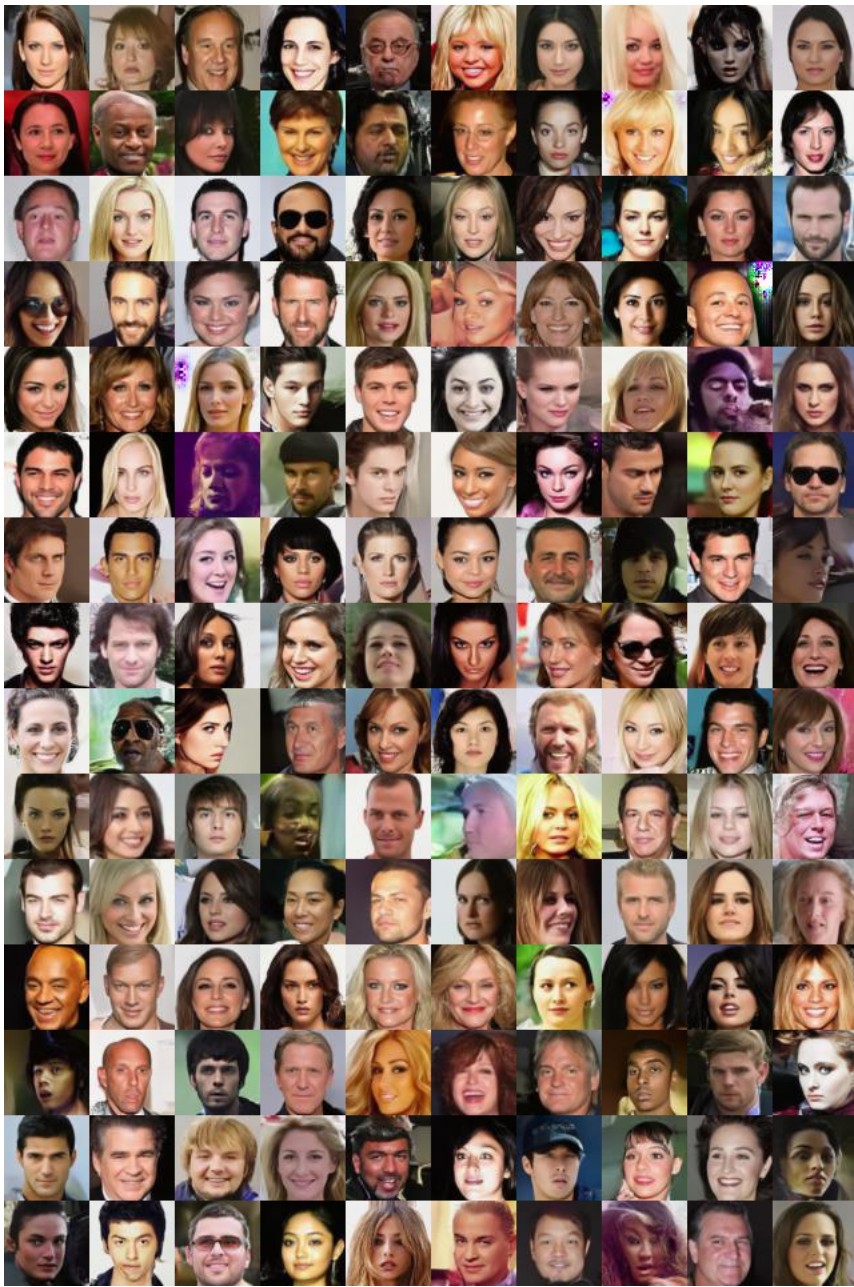

**Figure 8:** Additional samples from CelebA-64 at $t = 0.7$.

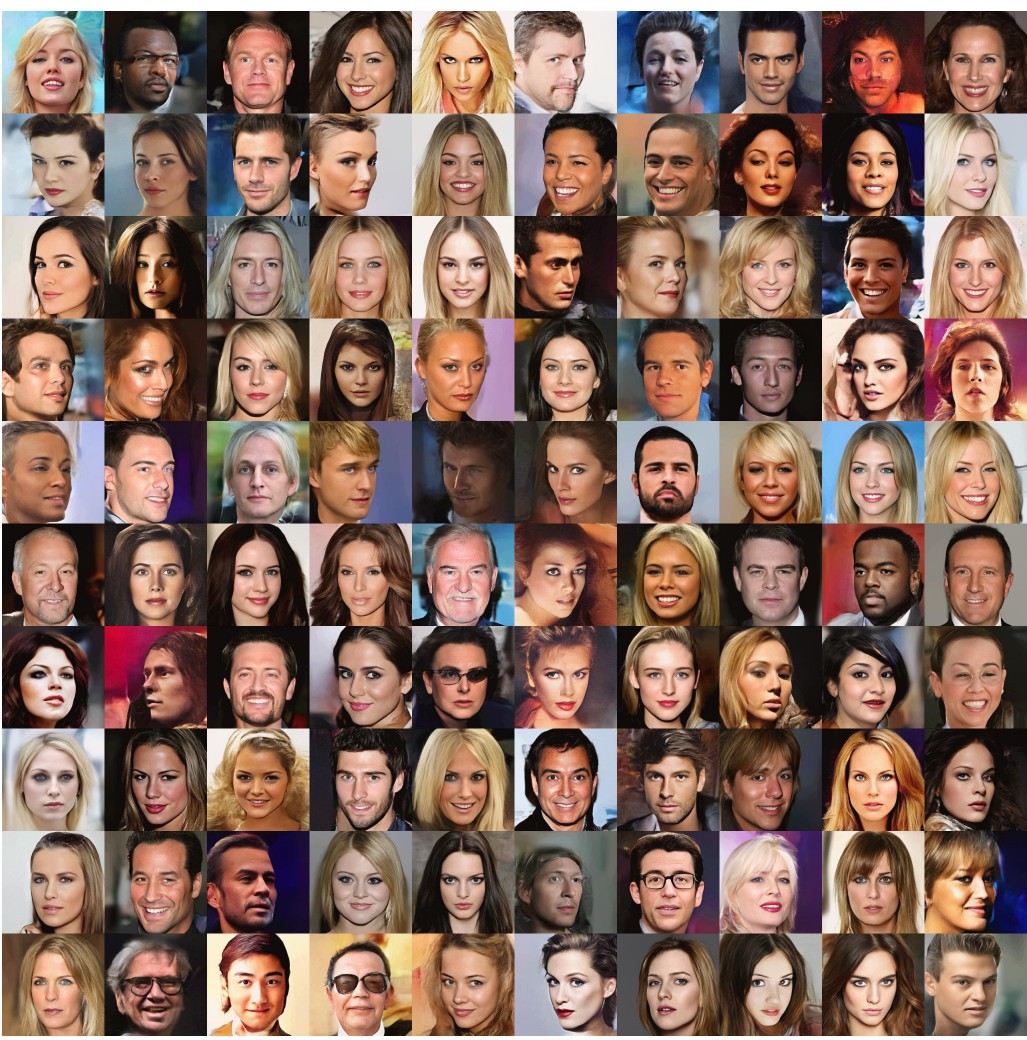

**Figure 9:** Additional samples from CelebA-HQ-256 at $t = 0.7$.

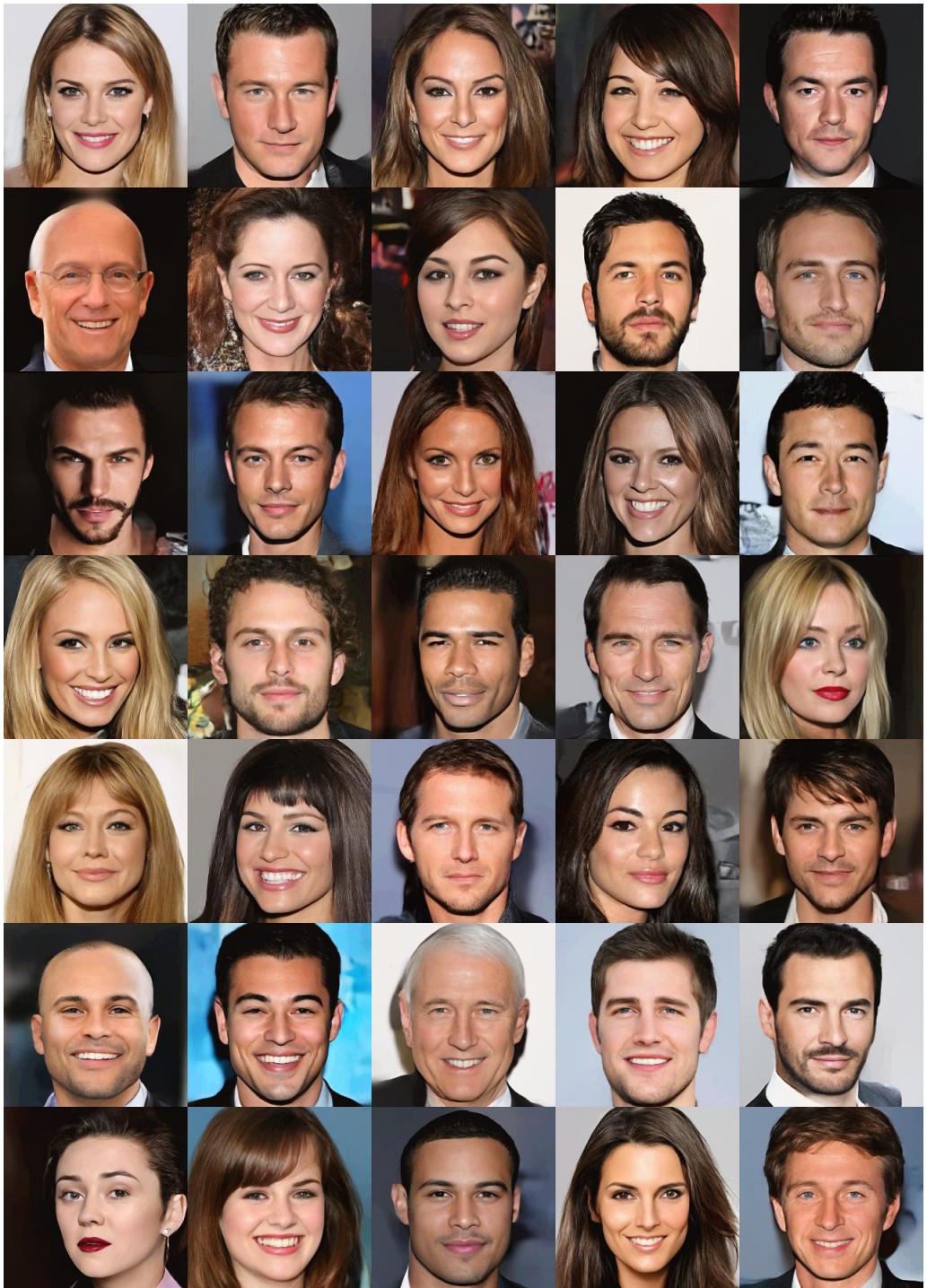

**Figure 10:** Selected good quality samples from CelebA-HQ-256.

## I EXPERIMENT ON SYNTHETIC DATA

In Fig. 11 we demonstrate the efficacy of our approach on the 25-Gaussians dataset, that is generated by a mixture of 25 two-dimensional Gaussian distributions that are arranged on a grid. The encoder and decoder of the VAE have 4 fully connected layers with 256 hidden units, with 20 dimensional latent variables. The discriminator has 4 fully connected layers with 256 hidden units. Note that the samples decoded from prior p(z) Fig. 11(b)) without the NCP approach generates many points from the the low density regions in the data distribution. These are removed by using our NCP approach (Fig. 11(c)).

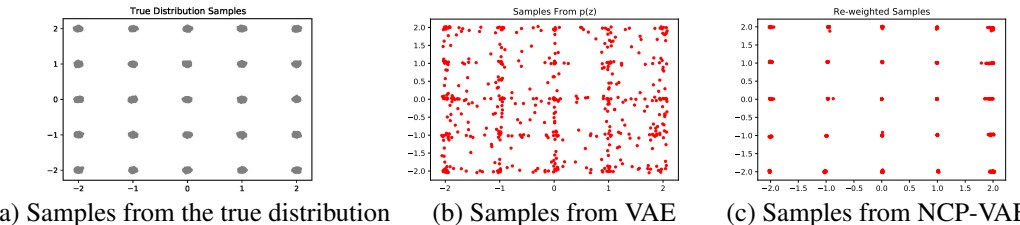

(a) Samples from the true distribution     (b) Samples from VAE     (c) Samples from NCP-VAE

**Figure 11:** Qualitative results on mixture of 25-Gaussians.

We use 50k samples from the true distribution to estimate the log-likelihood. Our NCP approach obtains an average log-likelihood of $-0.954$ nats compared to the log-likelihood obtained by sampling from the Gaussian prior, $-2.753$ nats. We use 20k Monte Carlo samples to estimate the log partition function for the calculation of log-likelihood.

## J    ADDITIONAL QUALITATIVE EXAMPLES

In Fig. 12, we show additional examples of images generated by NVAE (Vahdat & Kautz, 2020) and our NCP-VAE. We use temperature=0.7 for both. Visually corrupt images are highlighted with a red square.

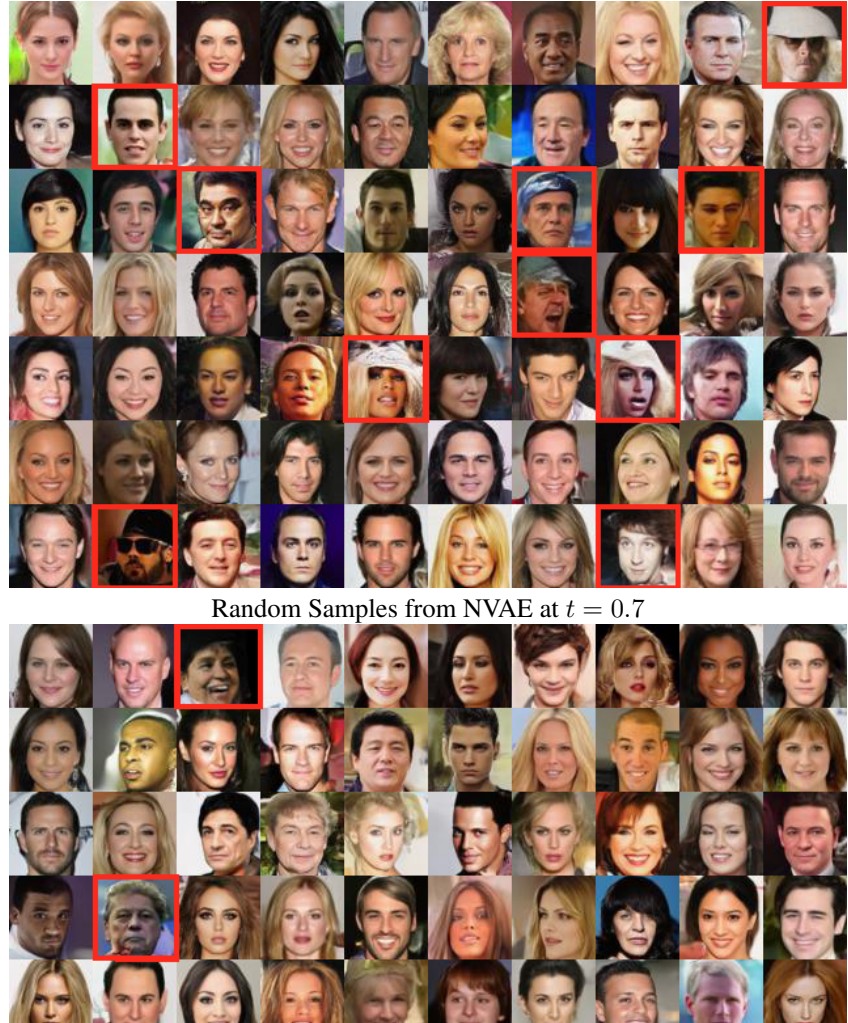

Random Samples from NVAE at $t = 0.7$

Random Samples from NCP-VAE at $t = 0.7$

**Figure 12:** Additional samples from CelebA-64 at $t = 0.7$.

