# OpenReview forum: "NCP-VAE: Variational Autoencoders with Noise Contrastive Priors"
_ICLR.cc/2021/Conference — Reject_

### Official Review · AnonReviewer4 · 2020-10-27
**Issues with experimental setup**

**Rating:** 5
**Confidence:** 3

**Review:**

Authors approach the "hole problem" of variational autoencoders where the aggregate posterior fails to match the prior, causing some areas of the prior distribution to be left out. Consequently, the decoder is not trained properly to operate in such regions, and the whole generate models is then subject to suboptimal performance. To attack this problem authors introduce two changes:
- Once a standard VAE is trained, a post-hoc learning of the prior distribution is performed. This is expected to move the prior closer to the aggregate posterior.
- In order to increase prior's expressivity, authors employ Energy Based Models (EBMs). To sidestep the difficulties of the likelihood-based training of EBMs authors switch to Noise Contrastive Estimation (NCE) procedure. Two techniques are proposed to perform approximate sampling from the EBM.

**Strong points**:
The paper is clearly written, technically correct and presents a new state-of-the-art method that significantly improves sample quality of Variational Autoencoders.

**Weak points**:
My major problem with the paper is the seeming disparity between the models in the experimental comparison. In particular, authors build NCP-VAE upon NVAE [1], a state-of-the-art VAE architecture notorious for its large size and use of many stochastic layers (groups of latent variables in authors' terms). At the same time, for both multistage VAE baselines (*2 stage VAE* and *RAE*) the FID metrics were taken from the corresponding papers, which were based on much simpler architectures with a single group of latent variables.

A more fair comparison would be to take the NVAE architecture and apply techniques from *RAE* [2] / *2 stage VAE* [3] to it. In the end of the sec. 4 authors say "It is not clear how 2s-VAE or RAE are applied to state-of-the-art hierarchical models". I believe this is a doable task. For example, consider (perhaps appropriately downscaled) NVAE's encoder (fig. 2a from [1]) and replace the input x with a random noise ε – the resulting latent variable model could be used as a second stage VAE [3]. For the *RAE* remove the red bottom-up path completely and make distribution over z1 a mixture. It might also be interesting to consider a mixture of size 1 -- a single gaussian distribution. In this case the 2nd stage training would be moving the prior towards the aggregate posterior. While in theory one might expect p(z) already be a decent approximation for q(z) (as shown in appendix B), in practice suboptimality of stochastic joint optimization might prevent p(z) from matching the aggregated q(z) to the fullest. The NVAE shares some weights between the prior and the encoder, which could either alleviate this problem by ensuring both encoder and prior change in a similar fashion, or hurt prior's expressivity since the top-down path has to be able to work in two somewhat different scenarios.

Finally, the qualitative evaluation seems disconnected from the problem being solved. It is my understanding that lessening the prior hole problem should result in a fewer unappealing / blurry samples generated by the model. The additional samples in appendix H still contain some such samples. Unfortunately, no such samples are presented for the baseline models (in [1] the samples were curated). It's therefore hard to tell if the frequency of such samples was reduced.

**Conclusion**: the results presented in the paper are impressive and the quality of samples is indeed very high. Unfortunately, I have to vote for rejection of this paper in its current form for its lack of rigorous experimental evaluation. It feels like the authors did overly focus on presenting a new method and have not given the prior work their best effort.

My suggestions to authors:
- Evaluate two stage approach to the NVAE: 1) nested VAE prior as in [3], 2) mixture over z1 as in [2], 3) fine-tune NVAE's prior. The later will also disentangle the benefits of an energy-based prior from those of the two-stage training.
- Perform qualitative evaluation based on frequency of visually unappealing samples from NCP-VAE versus that of the NVAE.

[1]: Arash Vahdat and Jan Kautz.  NVAE: A deep hierarchical variational autoencoder.

[2]: Partha Ghosh, Mehdi S. M. Sajjadi, Antonio Vergari, Michael Black, and Bernhard Scholkopf.  From variational to deterministic autoencoders.

[3]: Bin Dai and David Wipf.  Diagnosing and enhancing vae models.



## Post-rebuttal update

Authors have addresses some of the issues outlined above, in particular the additional comparison in sec. 5.4 and table 7 is informative. However, the RAE numbers indicate that a simple 10-component Gaussian Mixture is superior to the complex model of NCP-VAE with a Gaussian prior on a small VAE, and the superiority of NCP with GMM prior base provides little information as RAE with 50-components GMM could have performed even better. It's also not clear how 2s-VAE, RAE and WAE would scale to bigger architectures such as NVAE, which raises the question if the NCP was the optimal one.

It's surprising to see the 2-stage VAE to perform worse than the standard VAE, whereas (Dai & Wipf, 2018) have shown a significant improvement in the original paper. I think, this result needs further elaboration and validation.

Regarding the qualitative evaluation: figure 12 does show that the NCP-NVAE has fever poor samples, although it's hard to tell whether the difference is significant. Another issue is that the comparison does not include prior works.

As a result I bump by score to 5, but I think the paper still needs more work to make a sound argument for the proposed idea.

---

> ### Author Response · Authors · 2020-11-24
> **Response to Reviewer 4**
>
> We thank the reviewer for the feedback.
>
> •	**Additional comparison**
>
> The reviewer suggests applying the models from 2s-VAE and RAE to NVAE. Applying these models to NVAE may require making design decisions for 2s-VAE and RAE that are not originally addressed in these works. Instead, we suggest applying our noise contrastive prior (NCP) to the RAE architecture [1]. Since RAE [1] is compared to 2s-VAE and other baselines, we can compare our method against both RAE and 2s-VAE on exactly the same architecture (see Table 1 in [1]).
>
> Our results are reported in Table 7 in the revised version. Here, we examine how our NCP-VAE performs when the base prior is i) a Gaussian distribution or ii) a Gaussian mixture model with 10 components as suggested by RAE [1]. Our NCP-VAE improves the performance on the base VAE, improving the FID score to 41.28 from 48.12. Additionally, when NCP is applied to the VAE with GMM prior (the RAE model), it improves its performance from 40.95 to the FID score of 39.00. This experiment demonstrates the efficacy of our model over RAE and 2s-VAE implemented on exactly the same base VAE architecture.
>
> •  **Qualitative Evaluation**
>
> For a qualitative comparison to NVAE, in Appendix-J, we provide random images sampled from NVAE and NCP-VAE at a temperature t = 0.7. Additionally, similar to experiments by Elfeki et al. [4], in Appendix-I, we also present results on synthetic 2D data, constructed using a mixture of Gaussians on a grid. Results show that our method doesn’t sample from low density regions in the true data distribution.
>
> &nbsp;
>
> [1] From Variational to Deterministic Autoencoders, Ghosh et al., https://arxiv.org/abs/1903.12436
>
> [2] GDPP: Learning Diverse Generations Using Determinantal Point Process, Elfeki et al., https://arxiv.org/abs/1812.00068

---

### Official Review · AnonReviewer1 · 2020-10-27
**A new energy-based marginal distribution over latents for VAEs**

**Rating:** 8
**Confidence:** 5

**Review:**

**GENERAL**
The goal of the paper is to model the marginal over latents in VAEs in such a way to minimize the mismatch with the aggregated posterior. The paper proposes a new class of marginal distributions over the latent space that is a product of two experts: the first expert is a non-trainable probability distribution, and the second expert is an unnormalized probability distribution parameterized using neural networks. Since training a product of experts requires to apply an approximate inference (e.g., MCMC sampling), the authors propose to use the likelihood ratio trick. Eventually, a VAE is trained in two stages. First, they assume the marginal over z's to be simply the non-trainable distribution, and the VAE is trained. At the second stage, they propose to train the second expert (i.e., the binary classifier that distinguishes z ~ q(z) and z ~ p(z)) in order to obtain the final NCP that better matches the aggregated posterior. Further, the idea is extended to hierarchical VAEs, and a separate binary classifier is trained per each stochastic level.

**Strengths:**
S1: The idea is very interesting and allows to enrich the marginal distribution over z's in the VAE framework.

S2: The paper is well positioned in current trends in generative modeling. I find the combination of energy-based models and VAEs as a very appealing research direction.

S3: The proposed two-stage learning procedure is very logical and seem to be efficient. Its simplicity increases its reproducibility.

S4: Obviously, introducing an energy-based component results in intractability of the likelihood function due to the partition function. However, the FID scores and the quality of generated images are very convincing.

**Deficiencies:**
D1: In this review, I kept using "a non-trainable expert" or "a non-trainable marginal" for the base prior (the term used by the authors). However, I am not quite sure whether the base prior is non-trainable. I was unable to find any information about it in the paper. Therefore, I assumed it is a standard Gaussian after inspecting Figure 1. It would be easier for a reader, if this information was included in the text.


**Remarks:**
R1: The Appendix B.1 is closely related to the following papers:
- Rezende, D. J., & Viola, F. (2018). Taming vaes. arXiv preprint arXiv:1810.00597.
- Matthew D Hoffman and Matthew J Johnson. Elbo surgery: yet another way to carve up the variational evidence lower bound. In Workshop in Advances in Approximate Bayesian Inference, NIPS, 2016.
- Jakub Tomczak and Max Welling. Vae with a vampprior. In International Conference on Artificial Intelli- gence and Statistics, pp. 1214–1223, 2018.
It is maybe worth to mention these papers there. Otherwise, the text may sound as a new contribution.

R2: Section 3, first paragraph: The authors stated that: "Recently, energy-based models have shown promising results in representing complex distributions". This statement is very misleading, because the energy-based models (e.g., Boltzmann machines) have been used in ML for over 30 years, e.g.:
- Ackley, D. H., Hinton, G. E., & Sejnowski, T. J. (1985). A learning algorithm for Boltzmann machines. Cognitive science, 9(1), 147-169.
- Hinton, G. E., & Sejnowski, T. J. (1986). Learning and relearning in Boltzmann machines. Parallel distributed processing: Explorations in the microstructure of cognition, 1(282-317), 2.
- Salakhutdinov, R., & Hinton, G. (2009, April). Deep boltzmann machines. In Artificial intelligence and statistics (pp. 448-455).
- Larochelle, H., & Bengio, Y. (2008, July). Classification using discriminative restricted Boltzmann machines. In Proceedings of the 25th international conference on Machine learning (pp. 536-543).

The word "recently" suggests that this is a new invention that is simply not true.

R3: The proposed prior (NCP) could be seen as a specific form of a product of experts (e.g., Hinton, G. E. (2002). Training products of experts by minimizing contrastive divergence. Neural computation, 14(8), 177). In my opinion, it is an interesting connection.

R4: I wonder whether it is feasible to use some sort of importance sampling (e.g., Annealed Importance Sampling, Salakhutdinov, R., & Murray, I. (2008, July). On the quantitative analysis of deep belief networks. In Proceedings of the 25th international conference on Machine learning (pp. 872-879).) or other procedure (e.g., Perturb-and-MAP, Hazan, T., & Jaakkola, T. (2012, June). On the partition function and random maximum a-posteriori perturbations. In Proceedings of the 29th International Conference on International Conference on Machine Learning (pp. 1667-1674).) to estimate the partition function.

**AFTER REBUTTAL**
I would like to thank the authors for their rebuttal and all updates. The paper is related to other ideas in the literature, however, it constitutes an interesting contribution to the field. I appreciate all new results and discussions. I keep my original score.

---

> ### Comment · AnonReviewer1 · 2020-11-23
> **No rebuttal.**
>
> I do not see any rebuttal. However, I still believe the paper is really good even though similar ideas were published before.

---

> > ### Comment · Area_Chair1 · 2020-11-23
> > **RE:**
> >
> > R1, can you clarify your reasoning? The relevant prior work surfaced by other reviewers and lack of systematic comparison seems to substantially diminish the strength of this work.

---

> > > ### Author Response · Authors · 2020-11-23
> > > **Response document is coming soon**
> > >
> > > Dear reviewer and AC,
> > >
> > > We are in the processing of submitting a revised version with additional experiments, addressing the reviewers' comments. We apologize for the delay and we do hope that these additional experiments will demonstrate the efficacy of our proposed model compared to the prior arts.
> > >
> > > Thanks,

---

> > > > ### Comment · AnonReviewer1 · 2020-11-24
> > > > **After the rebuttal**
> > > >
> > > > Dear AC and the authors,
> > > >
> > > > I am really glad to see the rebuttal. I find the responses sensible and to the point. In my opinion, the authors significantly improved the quality of the paper by including new results, adding missing references and discussions.
> > > >
> > > > I am keeping the original score.
> > > >
> > > > Best.

---

> ### Author Response · Authors · 2020-11-24
> **Response to Reviewer 1**
>
> We thank the reviewer for the positive and encouraging feedback.
>
> •	**Discussion in Sec B.1**
>
>  Thanks for pointing this out. We added a short discussion to Sec. B to clarify that the derivation in this section is not a new contribution.
>
> •	**Energy-based models**
>
> The word “recently” in the description of prior work on EBM was meant for neural network based energy functions. We agree with you that EBMs have a rich history in machine learning and physics. We added an additional discussion to Sec. 4 to point out early work in this space.
>
> •	**Importance sampling and AIS for partition function estimation**
>
>  The main issue with using importance sampling or AIS for partition function estimation is that they provide stochastic lower bounds on $log Z$. Since $log Z$ participates in the log-likelihood expression with a negative sign, these methods tend to overestimate the log-likelihood. This makes a comparison to VAEs unfair as they are usually examined with lower bounds on their log-likelihood. Moreover, $log Z$ estimation is more challenging for EBMs defined in a large space. For example, Du & Mordatch NeurIPS 2019 [1] observed that AIS could not reliably estimate $log Z$ of an EBM defined on CIFAR-10 even after 2 days.
> We provide log-likelihood estimates for NCPs that are defined in a very small ‘hierarchical’ latent space in Tab. 4. Additionally we provide a comparison to other ‘non hierarchical’ models with a single group implementation in Tab. 8. We intentionally kept the latent space very small so that our $log Z$ estimates are reliable.
>
> •	**Gaussian Prior**
>
> Thank you for pointing this out. We indeed use a Gaussian prior. We clarified this in Sec. 3.4.
>
> &nbsp;
>
> [1] Implicit Generation and Generalization in Energy-Based Models, Du & Mordatch, https://arxiv.org/abs/1903.08689

---

### Official Review · AnonReviewer3 · 2020-10-29
**Novelity is unclear and additional experimental evidence needed**

**Rating:** 5
**Confidence:** 4

**Review:**

The main concerns are,
* The idea is somewhat novel but very similar to ideas developed in  [1,2] and (Bauer & Mnih, 2019). Note that, in [1] rejection sampling is performed on samples from the prior distribution based on the likelihood ratio obtained from a discriminator. While the results reported in [1] are not competitive to those reported in this work, it is probably be due to differences in model architecture. Therefore, a fair comparison to prior works [1,2] and (Bauer & Mnih, 2019) is required using a common model architecture. E.g. experiments using a simple DCGAN style architecture on CIFAR-10  would be sufficient to indicate efficacy.

* Prior resampling vs more complex priors: While the proposed NCP sampling scheme is applied on a Gaussian base prior with the NVAE architecture, it is unclear whether a similar gain in performance can be obtained using a more complex prior e.g. flow [3] or autoregressive (Van Den Oord et al., 2017; Razavi et al., 2019), [4]. A simple experiment which could help here is: keeping the architecture constant compare NCP with Gaussian base vs a flow or autoregressive prior.

* Additionally, no comparison to state of the art VAE based image synthesis approaches like VQ-VAE-2 (Razavi et al., 2019) are provided e.g. on ImageNet 256 × 256.

* No comparison to VAE-GAN based approaches [5,6], which achieves FIDs of  23.4 (vs 24.08 of this paper). It it also unclear whether the proposed approach can be applied on top of VAE-GAN based approaches.

* The current work uses the proposed NCP sampling scheme on a Gaussian base prior. Can the proposed prior be applied to more complex priors e.g. flow or autoregressive priors?

* No qualitative comparison of sample quality: Figure 3 only shows samples for the NCP approach. The paper should qualitative demonstrate that NCP leads to reduction in the number of "corrupted" images. To do this one possibility would be to demonstrate that the worst sample image in a batch from the NCP approach is on average significantly better than that from the plain NVAE approach. Alternatively, an even simpler experiment can be performed on the 2d Ring and Grid [7] datasets to demonstrate gains in sample quality.

[1] Dual Rejection Sampling for Wasserstein Auto-Encoders, ECAI 2020.

[2] Variational Rejection Sampling, AISTATS 2018.

[3] Latent Normalizing Flows for Many-to-Many Cross-Domain Mappings, ICLR 2020.

[4] Normalizing Flows with Multi-scale Autoregressive Priors, CVPR 2020.

[5] Variational Approaches for Auto-Encoding Generative Adversarial Networks, arXiv 2017.

[6] "Best-of-Many-Samples" Distribution Matching, NeurIPS Workshop 2019.

[7] GDPP: Learning Diverse Generations using Determinantal Point Process, ICML 2019.

---

> ### Author Response · Authors · 2020-11-24
> **Response to Reviewer 3**
>
> We thank the reviewer for the feedback.
>
> •	**Comparison to Prior Work - Non-hierarchical latent variable models**
>
> For a fair comparison to gauge the efficacy of our approach on non-hierarchical models, We apply NCP to commonly used small VAE models. In Tab. 7, and Tab. 8 we provide FID and log-likelihood respectively.
>
> 1.	Log-Likelihood: For a fair comparison with non-hierarchical approaches in Tab. 8 we compare log-likelihood of our NCP approach applied to the architecture of Lawson et al [4]. Our NCP approach applied to a Gaussian prior results in a negative log-likelihood (NLL) of 82.82 which is comparable to that of [4], i.e., 82.52. Note, that our approach results in a better NLL when compared to that obtained from using a Gaussian prior (84.82) and also the non-hierarchical methods of Bauer and Mnih [5] (83.03)
> However, for other datasets, we provided FID scores which are commonly used when log-likelihood is not available.
>
>
> 2.	FID: In Tab. 7 We show the generative performance of our approach applied to the VAE architecture in RAE by Gosh et al. [1]. Note that this VAE architecture has only one latent variable group. The same base architecture was used in the implementation of 2s-VAE by Dai and Wipf [6] and WAE by Tolstikhin et al. [7]. We borrow the training setup from RAE [1] on the CelebA-64 dataset, and compare to the baselines reported in this work. We apply our NCP-VAE on top of vanilla VAE with a Gaussian prior as well as a 10-component Gaussian mixture model (GMM) prior that was proposed in RAEs. Our NCP-VAE improves the performance on the base VAE, improving the FID score to 41.28 from 48.12. Additionally, when NCP is applied to the VAE with GMM prior (the RAE model), it improves its performance to an FID of 39.00.
>
> •	**Comparison to VQ-VAE**
>
>  VQ-VAE uses powerful autoregressive PixelCNN priors, applied to latent variables as large as 128x128 dimensions (in VQ-VAE v2). We expect the sampling speed from VQ-VAE to be very slow.
>
> •	**Comparison to VAE-GAN**
>
>  Our main focus is to improve the expressivity of the VAE prior through simple energy-based models. We believe that it is unfair to compare our method against VAE-GAN models that apply adversarial training in the data space.
>
> •	**More Complex Priors**
>
> 1.	Normalizing Flows applied to prior:  Note that section 3.2 by Che et al. [2] shows that applying a normalizing flow on the prior is equivalent to applying its inverse to the approximate posterior. Given that NVAE uses normalizing flows in its approximate posterior, we don’t believe adding additional normalizing flows increases the expressivity of the prior. We believe EBMs may provide additional expressivity that is not available through normalizing flows.
>
>
> 2.	Autoregressive Priors: Using autoregressive latent spaces following the spirit of PixelCNN could be applied. But note that these approaches tend to be very slow during sampling.
>
>
> •	**Qualitative comparison/Synthetic Example**
>
> For a qualitative comparison to NVAE we provide random images sampled from NVAE and NCP-VAE at a temperature t = 0.7. We specifically point out the corrupt images for both the methods in red in Appendix-J. Additionally, following the experiments of Elfeki et al. [4], we also perform experiments on 2D synthetic data using a mixture of Gaussians on a grid. Appendix-I shows the efficacy of our method in avoiding to generate samples from the low density regions in the true data distribution.
>
> &nbsp;
>
> [1] From Variational to Deterministic Autoencoders, Ghosh et al.,  https://arxiv.org/abs/1903.12436
>
> [2] Variational Lossy Autoencoder, Che et al., https://arxiv.org/abs/1611.02731
>
> [3] Resampled Priors for Variational Autoencoders, Bauer and Mnih,  https://arxiv.org/abs/1810.11428
>
> [4] GDPP: Learning Diverse Generations Using Determinantal Point Process. Elfeki et al., https://arxiv.org/abs/1812.00068
>
> [5] Energy-Inspired Models: Learning with Sampler-Induced Distributions, Lawson et al., https://arxiv.org/abs/1910.14265

---

### Official Review · AnonReviewer2 · 2020-11-03

**Rating:** 6
**Confidence:** 4

**Review:**

Summary:

The authors highlight an important problem in VAE - the prior-hole problem - which is that the approximate posterior and the simple gaussian prior do not match in spite of the KL term in the ELBO which makes sampling an issue - leading to the prior putting probability mass on latents that are not decoded to high probability mass regions in data manifold. Prior approaches have overcome this problem by increasing the expressivity of the prior through autoregressive models, and/or using hierarchical latents, EBMs with MCMC sampling. This paper proposes a very simple two stage method - (1) train a regular VAE, (2) train a binary classifier in NCE style to distinguish samples from prior and approx. posterior; use the re-weighting term from the NCE score to sample from a better re-weighted prior - either through langevin dynamics or re-sampling. The authors combine this approach with the use of hierarchical latents and produce really good performing generative models on a host of benchmarks with good looking samples.

Pros:

very simple and neat approach that produces really good results.
looks easy enough to reimplement and adopt widely for future research in VAEs.
samples look great, FID scores are good.
ablations on LD and SIR are very useful.
Cons:

would have been nice not to have a 2-stage approach. figuring out a way to have an online way of trying to train the re-weighting classifier in a GAN-like manner would be much more preferred.
paper can do a better job at citing related work - ex - should cite Variational Lossy Autoencoder - Chen et al 2016 - on using autoregressive priors and pointing out a connection to IAF posterior.
would be nice to show results even w/o hierarchical latents - how much improvement can be obtained over vanilla deep VAE with this two-stage approach
report results on log-likelihood benchmarks - CIFAR10, ImageNet-32, ImageNet-64 - compare to Flow models, Autoregressive Models, etc.

Post Rebuttal:
I believe the positioning of the paper could be improved but at the same time empirical results in the paper are strong. I am adjusting my score to 6 with a confidence of 4.

---

> ### Author Response · Authors · 2020-11-24
> **Response to Reviewer 2**
>
> We thank the reviewer for the positive feedback.
>
> *  **Two-stage Approach**
>
> Our two stage training approach is simple to train. Moreover, it can be easily applied to hierarchical models, as the noise contrastive training is done in parallel for all the groups. With recent studies showing the efficacy of very deep hierarchical VAEs [3], we believe that the appealing scalability of NCP allows its adoption to large hierarchical models.
>
> * **Citing Related Work**
>
> Thank you for the suggestion to cite “Variational Lossy Autoencoder”. We have added the citation in the introduction.
>
> * **Using complex priors**
>
> 1. Normalizing flows: Note that Sec. 3.2 by Che et al. [2] shows that applying a normalizing flow on the prior is equivalent to applying its inverse to the approximate posterior. Given that NVAE comes with normalizing flows in its approximate posterior, we don’t believe adding additional normalizing flows increases the expressivity of the prior. We believe EBMs may provide additional expressivity that is not available through normalizing flows.
>
>
> 2. Autoregressive models: Using autoregressive latent spaces following the spirit of PixelCNN could be applied. But note that these approaches tend to be very slow during sampling.
>
> * **Non-hierarchical latent variable models**
>
> For a fair comparison to gauge the efficacy of our approach on non-hierarchical models, in Tab. 7 and Tab. 8 we provide FID and log-likelihood respectively. We apply NCP to commonly used small VAE models.
>
> 1.	Log-Likelihood: For a fair comparison with non-hierarchical approaches, in Tab. 8, we compare log-likelihood of our NCP approach applied to the architecture of Lawson et al. [4]. Our NCP approach applied to a Gaussian prior results in a negative log-likelihood (NLL) of 82.82 which is comparable to that of [4], i.e., 82.52. Note that our approach results in a better NLL when compared to that obtained from using a Gaussian prior (84.82) and also the non-hierarchical methods of Bauer and Mnih [5] (83.03).
> However for other datasets, we provided FID scores which are commonly used when the log-likelihood is not available.
>
>
> 2.	FID: In Tab. 7 we show the generative performance of our approach applied to the VAE architecture of RAE by Gosh et al. [1]. Note that this VAE architecture has only one latent variable group. The same base architecture was used in the implementation of 2s-VAE by Dai and Wipf [6] and WAE by Tolstikhin et al. [7]. We borrow the training setup from RAE [1] on the CelebA-64 dataset, and compare to the baselines reported in this work. We apply our NCP-VAE on top of a vanilla VAE with a Gaussian prior as well as a 10-component Gaussian mixture model (GMM) prior that was proposed by Gosh et al. [1]. Our NCP-VAE improves the performance on the base VAE, improving the FID score to 41.28 from 48.12. Additionally, when NCP is applied to the VAE with GMM prior (the RAE model), it improves its performance to an FID of 39.00.
>
> &nbsp;
>
>
> [1] From Variational to Deterministic Autoencoders, Ghosh et al.,  https://arxiv.org/abs/1903.12436
>
> [2] Variational Lossy Autoencoder, Che et al., https://arxiv.org/abs/1611.02731
>
> [3] Very Deep VAEs Generalize Autoregressive Models and Can Outperform Them on Images, Anonymous submission, ICLR 2020.
> https://openreview.net/forum?id=RLRXCV6DbEJ
>
> [4] Energy-Inspired Models: Learning with Sampler-Induced Distributions, Lawson et al., https://arxiv.org/abs/1910.14265
>
> [5] Resampled Priors for Variational Autoencoders, Bauer and Mnih,  https://arxiv.org/abs/1810.11428
>
> [6] Diagnosing and Enhancing Vae Models, Dai and Wipf, https://arxiv.org/abs/1903.05789
>
> [7] Wasserstein auto-encoders, Tolstikhin et al., https://arxiv.org/abs/1711.01558

---

### Comment · Area_Chair1 · 2020-11-18
**Comments from the authors of a relevant prior work**

I asked the authors of Energy-Inspired Models: Learning with Sampler-Induced Distributions (Lawson et al. 2019) to comment on this paper. This was their response,

"Lawson, John, et al. "Energy-Inspired Models: Learning with Sampler-Induced Distributions." Advances in Neural Information Processing Systems. 2019.  presents a very similar scheme to this paper, including a method for learning to re-weight the prior of a VAE and training it using a NCE-esque objective (see section 3.2 "Self Normalized Importance Sampling"). I think that the authors should consider this paper for prior work.

The authors and reviewers may find Lawson et al. interesting because it considers aspects left unexplored by this work, including reweighting the output distribution in addition to the prior and training the entire system end-to-end instead of the two-step approach used here. It also situates this approach in a wider context of using sampling algorithms (in this case importance sampling) within generative models."

---

> ### Author Response · Authors · 2020-11-24
> **Positioning NCP-VAE Compared to the Prior Work**
>
> Thank you for pointing out the connection to SNIS [1]. Note that Lawson et al. [1] introduce energy-inspired models (EIMs) that define distributions induced by sampling processes (used by Bauer and Mnih [2] as well as our SIR sampling). Although EIMs have the advantage of end-to-end training and can be used as either a prior or decoder in VAEs, they require multiple samples during training (up to 1K samples). This makes the application of EIMs to deep hierarchical models such as NVAEs very challenging as these models are often very memory intensive and are trained with a few training samples per GPU.
>
> Our proposed two-stage training approach is simple to train. And, it can be easily applied to hierarchical models, as our noise contrastive training is done in parallel for all the groups. With recent studies showing the efficacy of very deep hierarchical VAEs [3], we believe that the appealing scalability of NCP-VAE permits its adoption to large hierarchical models.
>
> To ensure that we treat the prior art properly, we added a discussion on Lawson et al. [1] to our related work section. In our newly added Table 8, we provide an empirical comparison to both [1] and [2] using architectures/hyperparameters used in these works. We also added Table 7 which provides an additional comparison to several other works (suggested by the reviewers) using identical architectures/hyperparameters. We do hope that these additional experiments position our NCP-VAE better w.r.t. prior work.
>
> &nbsp;
>
> [1] Energy-Inspired Models: Learning with Sampler-Induced Distributions, Lawson et al., https://arxiv.org/abs/1910.14265
>
> [2] Resampled Priors for Variational Autoencoders, Bauer and Mnih,  https://arxiv.org/abs/1810.11428
>
> [3] Very Deep VAEs Generalize Autoregressive Models and Can Outperform Them on Images, Anonymous submission, ICLR 2020.
> https://openreview.net/forum?id=RLRXCV6DbEJ

---

### Author Response · Authors · 2020-11-24
**Responses Added and Paper Updated**

Dear reviewers,

We thank all of you for your detailed comments and feedback. We have submitted the response to all the reviews. Additionally, we provide an updated draft of the paper. The draft contains the following updates:

1. Section 5.4 (Tables 7 & 8), which includes an additional comparison to prior works. We majorly address the concern of comparisons with prior works, especially with non-hierarchical models. We provide FID scores for the CelebA-64 dataset and the negative log-likelihood for MNIST. For reference, we also provide the tables below.

2. Also, in Appendix-I we provide a new qualitative comparison using a synthetic 2d dataset of a 25 Gaussian Mixture on a grid, demonstrating that our method avoids sampling from the low-density region in the original data distribution.

3. Finally, Appendix J contains a new qualitative comparison of random samples from NVAE and our NCP-VAE approach. We demonstrate that random samples generated by our method contain a smaller number of corrupt images.

&nbsp;


| Model                         | FID   |
|-------------------------------|-------|
| VAE w/ Gaussian Prior         | 48.12 |
| 2s-VAE (Dai & Wipf, 2018)     | 49.70 |
| WAE (Tolstikhin et al., 2018) | 42.73 |
| RAE (Ghosh et al., 2020)      | 40.95 |
| NCP w/ Gaussian prior as base | 41.28 |
| NCP w/ GMM prior as base      | 39.00 |

**Table 7 from the main paper**
(Generative performance on CelebA-64 with the RAE (Ghosh et al., 2020) architecture)


&nbsp;
&nbsp;


| Model                                   | NLL   |
|-----------------------------------------|-------|
| VAE w/ Gaussian Prior                   | 84.82 |
| VAE w/ LARS prior (Bauer & Mnih, 2019)  | 83.03 |
| VAE w/ SNIS prior (Lawson et al., 2019) | 82.52 |
| NCP-VAE                                 | 82.82 |

**Table 8 from the main paper**
(Likelihood results on MNIST on single latent group model)

We hope that our responses have addressed your concerns. We will be glad to answer any of your concerns and questions. Thank you!


Authors of submission 959

---

### Decision · Program_Chairs · 2021-01-07
**Final Decision**

**Decision:**

Reject

**Comment:**

This paper is rejected.

All of the reviewers found the empirical results strong. However, R3 and R4 pointed out concerns with the positioning of the work relative to prior work and that their approach is conceptually similar to previous work. The authors have tried to address these concerns in their rebuttal. Both reviewers appreciate the changes, but still have remaining concerns that I agree with. Based on these concerns, it is unclear if the strong empirical results are mostly due to using the NVAE architecture, rather than a methodological improvement over previous methods. The authors should work on positioning their paper in the context of prior work and the comparisons requested by R3 and R4 for a resubmission.